# FORESTPERSONS: A LARGE-SCALE DATASET FOR UNDER-CANOPY MISSING PERSON DETECTION

**Deokyun Kim**[1]* **Jeongjun Lee**[2]* **Jungwon Choi**[2]* **Jonggeon Park**[2]*
Giyoung Lee[1]    Yookyung Kim[1]    Myungseok Ki[1]    Juho Lee[2]    Jihun Cha[1]

[1] Autonomous UAV Research Section, ETRI    [2] Kim Jaechul Graduate School of AI, KAIST
`{deokyunkim, giyoung, yk.kim, serdong, jihun}@etri.re.kr`
`{lee.jeongjun, jungwon.choi, parkjonggeon, juholee}@kaist.ac.kr`

## ABSTRACT

Detecting missing persons in forest environments remains a challenge, as dense canopy cover often conceals individuals from detection in top-down or oblique aerial imagery typically captured by Unmanned Aerial Vehicles (UAVs). While UAVs are effective for covering large, inaccessible areas, their aerial perspectives often miss critical visual cues beneath the forest canopy. This limitation underscores the need for under-canopy perspectives better suited for detecting missing persons in such environments. To address this gap, we introduce ForestPersons, a novel large-scale dataset specifically designed for under-canopy person detection. ForestPersons contains 96,482 images and 204,078 annotations collected under diverse environmental and temporal conditions. Each annotation includes a bounding box, pose, and visibility label for occlusion-aware analysis. ForestPersons provides ground-level and low-altitude perspectives that closely reflect the visual conditions encountered by Micro Aerial Vehicles (MAVs) during forest Search and Rescue (SAR) missions. Our baseline evaluations reveal that standard object detection models, trained on prior large-scale object detection datasets or SAR-oriented datasets, show limited performance on ForestPersons. This indicates that prior benchmarks are not well aligned with the challenges of missing person detection under the forest canopy. We offer this benchmark to support advanced person detection capabilities in real-world SAR scenarios. The dataset is publicly available at https://huggingface.co/datasets/etri/ForestPersons.

## 1 INTRODUCTION

Unmanned Aerial Vehicles (UAVs) have been widely used in Search and Rescue (SAR) missions because they can quickly cover large open areas. While early UAVs relied on manual operation, advances in navigation, path planning, and flight control technologies have enabled fully autonomous missions. Furthermore, hardware miniaturization has led to the development of Micro Aerial Vehicles (MAVs), and improvements in Simultaneous Localization and Mapping (SLAM) technologies have made GPS-denied navigation possible (Liu et al., 2022; Bachrach et al., 2010). These developments have extended UAV operations to challenging forest environments with dense and scattered obstacles. Recent studies have demonstrated that UAVs can perform safe navigation (Laina et al., 2024; Hong et al., 2024), rapid path planning for exploration (Ren et al., 2025; Jarin-Lipschitz et al., 2022; Zhou et al., 2021), and mapping tasks (Lin & Sto, 2022; Kwon et al., 2024). Despite the growing feasibility of deploying MAVs in forested environments, detecting missing persons under dense canopies remains a fundamental challenge. Forests are environments where people are not typically present, and the abundance of vegetation causes significant and often unpredictable occlusions. Moreover, there is a lack of dedicated datasets targeting such under-canopy scenarios, limiting the ability of detection models to learn and generalize to these challenging conditions.

While several UAV-based datasets (Kundid Vasić & Papić, 2022; Broyles et al., 2022; Sambolek & Ivasic-Kos, 2021; Zhang et al., 2025) have been introduced to support SAR applications, most

---

*Equal contribution

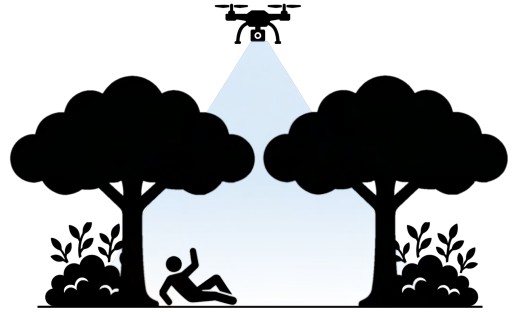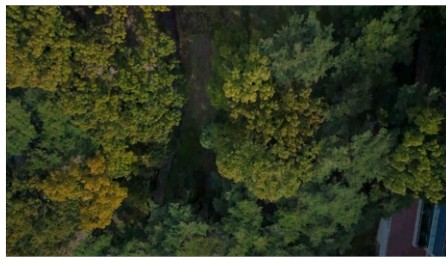

(a) High-altitude aerial UAV perspective: wide-area coverage but limited visibility under forest canopy.

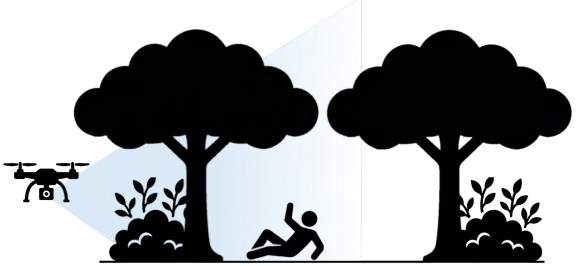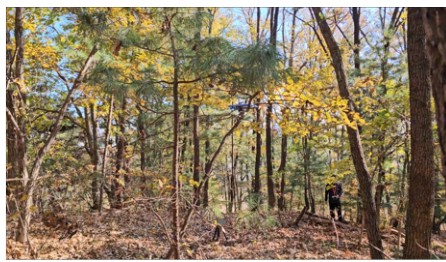

(b) Low-altitude MAV perspective: ground-level view under canopy with improved visibility of missing persons.

Figure 1: **Comparison of two UAV-based person search scenarios.** (a) High-altitude views offer wide-area coverage but often fail to detect targets due to canopy occlusion. (b) Low-altitude MAVs provide closer, ground-level views beneath the canopy, improving the chances of spotting missing persons despite vegetation occlusion.

prior benchmarks are collected from high altitudes, typically using top-down or oblique perspectives. Although such aerial viewpoints provide broad coverage and are effective for detecting objects in open areas, they are less suitable for locating missing persons concealed beneath dense forest canopy. At high altitudes, individuals often appear as only a few pixels in the image. Dense foliage and uneven terrain further obstruct visibility, making reliable detection extremely challenging. Moreover, occlusions caused by vegetation are pervasive and vary unpredictably across different forest structures, exacerbating the difficulty of identifying partially visible or collapsed individuals.

To address this challenge, we introduce **ForestPersons**, a large-scale dataset specifically designed to support the training of models for detecting missing persons under forest canopies, where dense vegetation often causes severe occlusion and obstructs the visibility of human bodies. The dataset consists of 96,482 images and 204,078 annotated instances, collected across varying seasonal, weather, and lighting conditions, reflecting real-world under-canopy scenarios. Each person instance is annotated with bounding boxes and additional attributes including pose and visibility, which are particularly relevant to SAR applications. To complement the RGB-only ForestPersons, we also publicly release a thermal IR image dataset, called **ForestPersonsIR**, and report its benchmark results in Appendix G. To the best of our knowledge, ForestPersons is the first benchmark explicitly designed for detecting persons under forest canopies, providing a foundation for developing and evaluating models in realistic SAR scenarios, and is expected to improve the chances of successful rescue of missing persons in real-world SAR missions.

## 2 RELATED WORK

### 2.1 UAV-BASED PERSON DETECTION DATASETS

Most prior UAV-based datasets capture people from top-down or oblique perspective at high altitudes as illustrated in Figure 1(a). Over the past several years, large-scale datasets (Zhu et al., 2021; Speth et al., 2022; Barekatain et al., 2017; Du et al., 2018; Zhu et al., 2022; Liu et al., 2023) containing high-resolution aerial imagery have been developed to support computer vision tasks

Table 1: **ForestPersons vs. Others.** Comparison of ForestPersons with existing UAV-based datasets containing person class annotations.

| Dataset | Configuration | | | Dataset Size | | Attributes | |
| | Scenario | Environments | View Point | #Images | #Annotations | Occlusion | Pose |
|---|---|---|---|---|---|---|---|
| HERIDAL (Kundid Vasić & Papić, 2022) | SAR | Forest | Top-down | 1,600 | 3,194 | ✗ | ✗ |
| WiSARD (Broyles et al., 2022) | SAR | Forest, Maritime | Oblique | 44,588 | 74,204 | ✗ | ✗ |
| SARD (Sambolek & Ivasic-Kos, 2021) | SAR | Forest | Oblique | 1,981 | 6,532 | ✗ | ✓ |
| VTSaR (Zhang et al., 2025) | SAR | Urban, Maritime, Forest | Top-down | 12,465 | 19,956 | ✗ | ✗ |
| Visdrone (Zhu et al., 2021) | Surveillance | Urban | Oblique | 10,209 | 147,747 | ✓ | ✗ |
| NII-CU (Speth et al., 2022) | Detection | Urban | Oblique | 5,880 | 18,736 | ✓ | ✗ |
| Okutama-Action (Barekatain et al., 2017) | Detection | Urban | Oblique | 77,365 | 524,649 | ✗ | ✓ |
| **ForestPersons** | **SAR** | **Forest** | **Ground-level** | **96,482** | **204,078** | ✓ | ✓ |

such as object detection, tracking, and person recognition from aerial perspectives. Among these, VisDrone dataset (Zhu et al., 2021) stands out as a comprehensive resource for drone-based computer vision applications, offering data captured using various drone-mounted cameras across diverse urban and country environments, locations, object types, and scene densities. Other notable general-purpose aerial datasets include NII-CU (Speth et al., 2022), which contains well-aligned RGB and thermal images with occlusion labels, and Okutama-Action (Barekatain et al., 2017), which provides aerial video for human action detection with bounding boxes and 12 action classes such as standing, sitting, and lying.

Several datasets have been proposed for various SAR applications. HERIDAL (Kundid Vasić & Papić, 2022) provides high-resolution imagery from mountainous regions, while WiSARD (Broyles et al., 2022) offers synchronized RGB and thermal data across diverse terrains and weather conditions. SARD (Sambolek & Ivasic-Kos, 2021) and the recently proposed VTSaR (Zhang et al., 2025) extend multimodal capabilities by incorporating real and synthetic RGB-thermal image pairs. Most UAV-based SAR datasets, however, are collected from high altitudes and primarily offer top-down or oblique viewpoints. While such perspectives are advantageous for efficiently covering wide areas, they are less effective in real SAR scenarios where missing persons are often located beneath dense foliage. In these environments, visibility is severely limited and occlusions caused by vegetation are frequent. As a result, this reduces the chances of successfully detecting missing persons in aerial imagery. Table 1 summarizes the key attributes of representative UAV-based detection datasets.

## 2.2 GROUND-LEVEL PERSON DETECTION DATASETS

As illustrated in Figure 1(b) MAVs typically operate at low altitudes close to ground-level view. Given the similarity in viewpoints, ground-level person detection datasets are suitable training resources for under-canopy missing person detection models. Representative prior works include COCO Lin et al. (2014), CrowdHuman (Shao et al., 2018), CityPersons (Zhang et al., 2017), KITTI (Geiger et al., 2012), and JRDB (Martin-Martin et al., 2021), which are widely used as benchmarks for developing and evaluating person detection models. These datasets provide high-resolution images captured in everyday environments, including annotations for bounding boxes, body joints, and occlusion states. They have supported the development of person detection models that are robust to partial occlusion and variations in human pose.

Nevertheless, most existing datasets primarily depict standing or walking individuals in typical indoor and outdoor environments where people are commonly found. These conditions differ substantially from those encountered in SAR missions conducted in forested environments. In real SAR scenarios, missing persons are often partially occluded by vegetation, sitting or lying beneath canopy cover, and subject to highly variable lighting and visibility conditions. Such characteristics are rarely captured in prior benchmarks, making existing datasets less suitable for training missing person detection models intended for under-canopy search operations.

## 3 FORESTPERSONS

ForestPersons is a large-scale image dataset specifically developed for missing person detection in under-canopy forest environments, a key task in autonomous SAR missions. The dataset captures conditions that are common in under-canopy forest searches, where people may be partially or fully

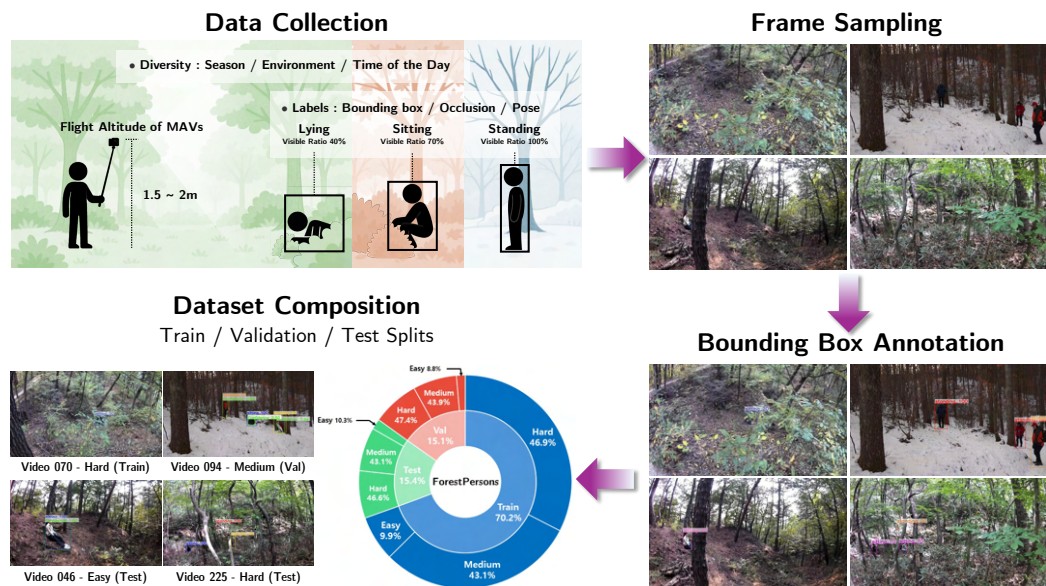

Figure 2: **Overview of ForestPersons composition pipeline.** The full process from data collection in forest environments to frame sampling from video sequences, bounding boxes annotation of missing persons, and difficulty-aware dataset splitting.

hidden by vegetation and can appear in various poses such as lying down, sitting, or standing. Unlike conventional person detection datasets that focus on images collected in places where people are typically found, ForestPersons targets under-canopy forest scenes, where dense foliage, seasonal shifts, and weather variability significantly impact visibility and scene appearance.

## 3.1 DATA COLLECTION AND FRAME SAMPLING

The ForestPersons dataset was constructed to simulate realistic SAR scenarios occurring under forest canopy conditions. As shown in Figure 2, videos were collected across diverse forest environments by simulating missing person situations that reflect plausible outcomes of fatigue or disorientation. Individuals were positioned in different postures such as lying on the ground, sitting, or standing. In these settings, they were naturally partially occluded by vegetation, branches, or uneven terrain. To emulate the viewpoints typically encountered by MAVs during under-canopy missions, handheld or tripod-mounted cameras were positioned at altitudes between 1.5 and 2.0 meters, approximating the expected flight height of MAVs.

The videos include scenes from different seasons, such as dense summer foliage that increases occlusion and winter settings with leafless trees and snow-covered terrain. Variations in weather, including clear skies, overcast conditions, and light rain, were incorporated to introduce changes in visibility and lighting. Temporal diversity was also considered by capturing footage at different times of day, primarily in the afternoon and at dusk. We deliberately included seasonal and temporal conditions in the videos to support the development of detection models that are robust to real-world SAR scenarios. Frames were extracted from the 377 video sequences collected as described above.

## 3.2 ANNOTATION

Bounding boxes were annotated using the open-source COCO Annotator (Brooks, 2019), following shared guidelines that required labeling only the visible portions of each individual. Given the dense vegetation and complex terrain characteristic of under-canopy environments, annotators were instructed to carefully delineate the visible contours of partially occluded individuals to ensure precise and consistent annotations.

In addition to bounding boxes, each person instance was annotated with two semantic attributes, pose and visibility level, to capture information relevant to practical SAR operations. The pose attribute

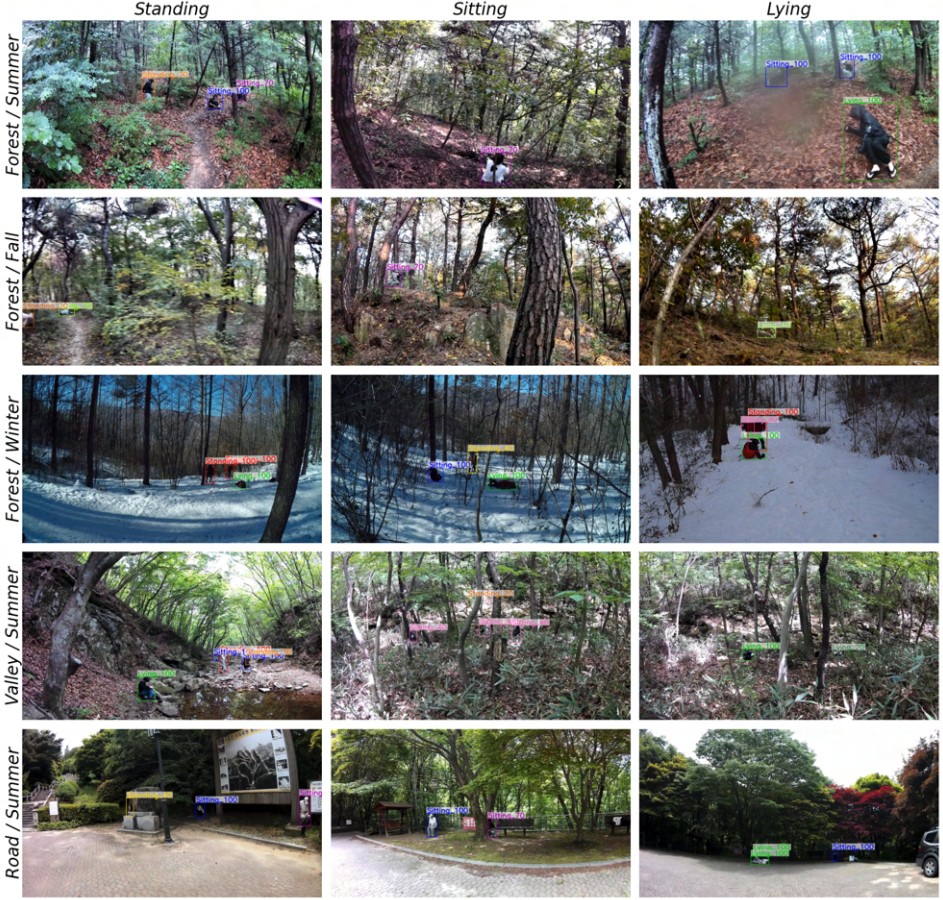

Figure 3: **Visual samples from ForestPersons.** Images depicting individuals in diverse poses, occlusion levels, seasons, and forest environments.

provides cues about the physical state of an individual, while visibility level quantifies the degree of visual difficulty caused by environmental occlusions. These interpretable categories are designed to reflect the visual conditions commonly encountered in real-world forest search scenarios.

Poses were categorized into three classes: standing, sitting, and lying. In cases where the posture of a person was ambiguous due to occlusion or background clutter, annotators referred to adjacent video frames to make informed decisions based on shared annotation guidelines. Visibility levels were categorized into four levels based on the degree of occlusion caused by vegetation or terrain: a value of 20 indicates heavy occlusion where the individual is almost unrecognizable, 40 corresponds to partial occlusion with the person still identifiable, 70 denotes minor occlusion with most of the body clearly visible, and 100 represents full visibility without any occlusion. Representative examples of each visibility level and pose category under realistic forest conditions are presented in Figure 3.

Following the annotation of bounding box and semantic attributes, an automated and manual anonymization protocol was applied to remove personally identifiable facial information. Specifically, a face detector (López, 2024) was used to identify facial regions in all images, which were then blurred accordingly. Subsequently, a manual review was conducted to identify any remaining visible faces, and additional blurring was applied as needed to ensure complete anonymization.

## 3.3 DATASET SPLIT AND STATISTICS

With the data collection and annotation processes described above, ForestPersons comprises 96,482 images and 204,078 annotated person instances, each instance labeled with a bounding box, pose, and visibility level. To reduce annotator bias and mitigate the effects of human error, we designed a model-driven difficulty-aware dataset splitting strategy. In particular, to prevent overlap between

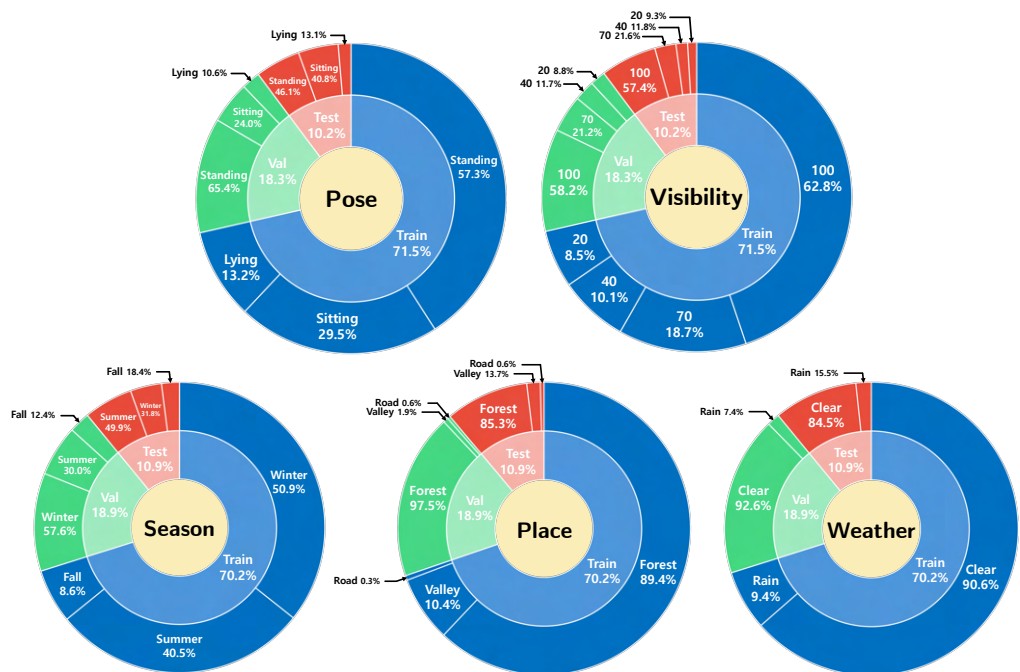

Figure 4: **Annotation statistics of ForestPersons.** Instance-level distribution for pose and visibility (Top) and image-level distribution for season, place, and weather (Bottom).

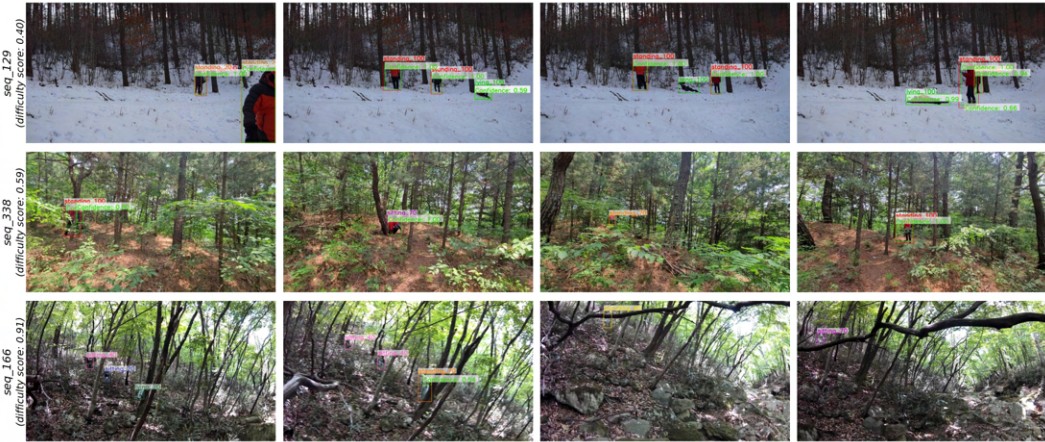

Figure 5: **ForestPersons samples by difficulty level.** Shown are representative video sequences from the easy, medium, and hard groups. Predicted boxes (green) are shown with confidence scores, and ground-truth boxes (other colors) are labeled as {*pose*}_{*visibility level*}.

temporally adjacent frames and to account for task difficulty, we split the dataset at the video sequence level. Each sequence was assigned a difficulty score based on the detection performance of a COCO-pretrained Faster R-CNN (Ren et al., 2015) implemented in Detectron2 (Wu et al., 2019), computed as $1 - \mathrm{AP}_{50}$. Sequences were then grouped such that easy, medium, and hard samples were proportionally distributed across the training, validation, and test splits, as detailed in Appendix E.

As shown in Figure 4, the training, validation, and test splits exhibit comparable distributions across seasons, location types, and weathers for images, as well as visibility levels and poses for the missing person instances. These distributions reflect biases from the image collection process, despite efforts to ensure scenario diversity. Nevertheless, each split maintains sufficient diversity to reflect real-world variability. To better simulate realistic SAR situations near forest entrances, a small number of videos recorded at forest edges (labeled as "Road") were also included in the dataset.

Representative examples from each difficulty group are shown in Figure 5, with one sample per row corresponding to easy (difficulty score $< 0.45$), medium ($0.45 \leq$ score $< 0.75$), and hard (score $\geq 0.75$) levels, respectively. The final split consists of 67,686 images and 145,816 annotations for training, 18,243 images and 37,395 annotations for validation, and 10,553 images and 20,867 annotations for testing.

# 4 EXPERIMENTS

## 4.1 EXPERIMENT SETTING

**Training object detection models.** We evaluate a diverse set of widely adopted and representative object detection models. Specifically, we train models with YOLO-based (Redmon et al., 2016) backbones (YOLOv3 (Redmon & Farhadi, 2018), YOLOX-tiny (Ge et al., 2021)) and YOLOv11-nano (Jocher & Qiu, 2024), ResNet-50-based (He et al., 2016) backbones (RetinaNet (Lin et al., 2017), Faster R-CNN (Ren et al., 2015)) and deformable Faster R-CNN (Dai et al., 2017), a MobileNetV2-based (Sandler et al., 2018) backbone (SSD (Liu et al., 2016)), and transformer-based (Vaswani et al., 2017) backbones (DETR (Carion et al., 2020) and DINO (Caron et al., 2021)). We also evaluate CZ Det (Meethal et al., 2023), a model designed for UAV imagery that utilizes a cascaded zoom-in mechanism. All models, except for YOLOv11, DINO, and CZ Det, are implemented using `MMDetection` framework (Chen et al., 2019). The YOLOv11, DINO and CZ Det is implemented using `ultralytics` (Jocher & Qiu, 2024), `detrex` (Ren et al., 2023), and `detectron2` framework (Wu et al., 2019), respectively. The training hyperparameters for each model are detailed in Table 6, Appendix B. We conduct all experiments on NVIDIA RTX 3090 GPUs, except for DETR models, which were trained on NVIDIA A100 and A6000 GPUs.

**Evaluation.** We use Average Precision (AP) and Average Recall (AR) as the primary evaluation metrics. Specifically, both are computed across Intersection over Union (IoU) thresholds ranging from 0.5 to 0.95 at intervals of 0.05. We report $AP_{50:95}$ as the main metric, along with $AP_{50}$ and $AP_{75}$, which correspond to IoU thresholds of 0.5 and 0.75, respectively. In SAR missions, where false negatives (i.e., missed detections of actual persons) can critically impact mission success, recall is especially important. We therefore report $AR_{50:95}$ to provide a complementary view of detection performance. We refer to $AP_{50:95}$ and $AR_{50:95}$ simply as AP and AR throughout the paper.

## 4.2 LIMITATIONS OF PRIOR DATASETS IN UNDER-CANOPY ENVIRONMENTS

Prior SAR datasets, which are composed of aerial imagery, present challenges for detecting persons under-canopy due to the difference in viewpoint and limited visibility caused by vegetation. Meanwhile, publicly available ground-level person datasets do not adequately account for occlusions caused by dense vegetation, making them less suitable for these tasks. To demonstrate this limitation, we conduct experiments to assess the generalization capability of models trained on these prior datasets when applied to our proposed dataset. Specifically, we train object detection models using existing SAR datasets and conventional ground-level person datasets, and evaluate their performance on the test split of ForestPersons.

Table 2: **Adaptation of prior datasets to under-canopy SAR tasks.** Performance comparison of Faster R-CNN trained on each dataset and evaluated on two test sets: the dataset's own test split and the ForestPersons' test split. (Left) UAV-based SAR datasets; (Right) ground-level person datasets.

| UAV-based SAR dataset | | | | | Ground-level person dataset | | | | |
|---|---|---|---|---|---|---|---|---|---|
| Train | Test | AP | $AP_{50}$ | $AP_{75}$ | Train | Test | AP | $AP_{50}$ | $AP_{75}$ |
| SARD | SARD | 58.6 | 90.8 | 68.4 | COCOPerson | COCOPerson | 54.0 | 82.5 | 58.2 |
| | ForestPersons | 3.0 | 7.8 | 1.6 | | ForestPersons | 40.8 | 66.9 | 45.2 |
| HERIDAL | HERIDAL | 35.0 | 70.8 | 29.3 | CrowdHuman | CrowdHuman | 39.4 | 74.8 | 37.3 |
| | ForestPersons | 0.2 | 0.3 | 0.2 | | ForestPersons | 31.9 | 58.8 | 31.0 |
| WiSARD | WiSARD | 18.5 | 51.7 | 7.9 | CityPersons | CityPersons | 38.7 | 62.5 | 42.1 |
| | ForestPersons | 11.3 | 29.0 | 6.4 | | ForestPersons | 5.9 | 15.1 | 3.7 |

The results, summarized in Table 2, indicate that models trained on UAV-based SAR dataset Lin et al. (2014); Shao et al. (2018); Zhang et al. (2017) performed poorly on ForestPersons, and those trained on ground-level dataset also showed significant performance degradation due to occlusions from natural elements in the forest, such as branches and foliage, and viewpoint differences, especially the aerial perspective common in SAR data. Meanwhile, models trained on ground-level person datasets struggle with individuals who are partially occluded by vegetation or in non-standing poses such as sitting or lying. These findings highlight the limitations of relying solely on existing SAR and ground-level datasets for under-canopy SAR applications, thereby underscoring the necessity and relevance of our proposed dataset. The examples of failure cases of the object detection models trained with existing datasets are depicted in Figure 9 in the Appendix.

## 4.3 DATASET BENCHMARK PERFORMANCE

Table 3: **ForestPersons benchmark results.** Object detection model performance on validation and test splits of ForestPersons.

| Detection Model | Validation Split | | | | Test Split | | | |
|---|---|---|---|---|---|---|---|---|
| | AP | $AP_{50}$ | $AP_{75}$ | AR | AP | $AP_{50}$ | $AP_{75}$ | AR |
| YOLOv3 | 55.6 | 91.7 | 63.2 | 63.1 | 50.2 | 86.5 | 53.9 | 58.6 |
| YOLOX | 56.8 | 92.9 | 65.2 | 62.5 | 51.0 | 89.0 | 54.4 | 58.2 |
| YOLOv11 | 65.3 | 95.4 | 76.6 | 71.5 | 65.6 | 93.4 | 75.6 | 71.7 |
| RetinaNet | 64.1 | 96.0 | 75.8 | 70.4 | 64.2 | 93.9 | 74.4 | 70.9 |
| Faster R-CNN | 64.2 | 95.6 | 76.5 | 69.6 | 64.4 | 92.7 | 75.4 | 70.0 |
| Deformable R-CNN | 65.0 | 94.7 | 78.5 | 70.0 | **66.3** | 93.4 | 77.5 | 71.3 |
| SSD | 48.9 | 88.5 | 49.4 | 57.8 | 45.0 | 83.6 | 43.1 | 53.7 |
| DETR | 55.3 | 93.0 | 59.9 | 68.0 | 53.9 | 88.7 | 59.4 | 67.9 |
| DINO | 59.9 | 91.7 | 69.1 | 70.1 | 65.3 | 94.0 | 76.2 | **77.7** |
| CZ Det | **69.9** | **98.1** | **83.4** | **76.8** | 65.6 | **96.1** | **77.9** | 71.6 |

We evaluated the baseline object detection models on ForestPersons, as summarized in Table 3. Our results show that YOLO-based models (YOLOv3, YOLOX, YOLOv11) achieve APs of 50.2, 51.0, and 65.6, respectively; ResNet-50-based detectors (RetinaNet, Faster R-CNN, Deformable R-CNN) obtain 64.2, 64.4, and 66.3; the MobileNetV2-based SSD records 45.0; Transformer-based models (DETR and DINO) reach 53.9 and 65.3; and CZ Det, incorporating a cascaded zoom-in mechanism for UAV imagery, achieves 65.6. While Deformable R-CNN attained the highest AP of 66.3, other models excelled in different key metrics. Specifically, DINO led in AR with 77.7, and CZ Det achieved the best scores for both $AP_{50}$ and $AP_{75}$, at 96.1 and 77.9, respectively. These results suggest that, for evaluating object detectors in SAR missions, ForestPersons highlights how different models excel under different evaluation criteria, making it possible to select methods according to mission-specific requirements.

## 4.4 IMPACT OF DIFFERENT ATTRIBUTES ON DETECTION PERFORMANCE

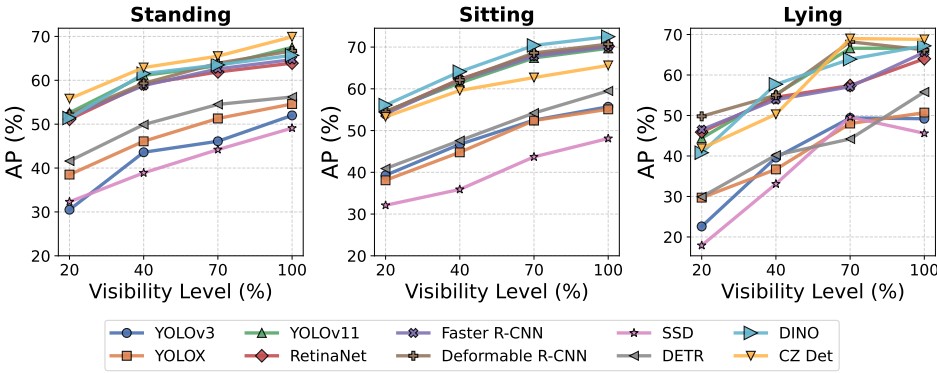

Figure 6: **Effect of visibility level on detection performance.** Detection precision improves as the visibility level increases across pose attributes.

**Visibility diversity reflecting real-world SAR conditions.** In under-canopy SAR tasks, it is natural that the difficulty of person detection increases as the degree of occlusion caused by surroundings becomes more severe. To simulate this challenge, ForestPersons includes human instances with varying levels of occlusion, which are carefully annotated with corresponding visibility level. Figure 6 shows that the performance of models trained on ForestPersons increases with the visibility level. The correlation between AP and visibility level empirically demonstrates the inherent difficulty of detecting heavily occluded individuals in under-canopy SAR tasks. The explicit annotation of pose and visibility level in ForestPersons enables systematic evaluation and facilitates the development of robust object detection models better suited for real-world SAR scenarios.

Table 4: **Impact of various attributes on detection performance in ForestPersons.** Each object detection model was trained and evaluated using subsets of train and test data with unique attributes.

| | (a) Pose | | | | | | (b) Season | | | | | | | | |
| --- | --- | --- | --- | --- | --- | --- | --- | --- | --- | --- | --- | --- | --- | --- | --- |
| Train Attributes | Standing | | | All Poses | | | Summer | | | Winter | | | All Seasons | | |
| Test Attributes | Standing | Sitting | Lying | Standing | Sitting | Lying | Summer | Fall | Winter | Summer | Fall | Winter | Summer | Fall | Winter |
| YOLOv3 | 45.3 | 30.0 | 32.1 | **49.3** | **51.5** | **47.5** | 49.7 | 53.7 | 25.7 | 4.5 | 1.4 | **54.0** | 51.1 | 58.2 | 50.7 |
| YOLOX | 47.3 | 30.3 | 31.7 | **52.2** | **50.6** | **47.9** | 56.8 | 57.1 | 17.2 | 5.5 | 1.5 | **60.0** | 50.0 | 53.6 | 56.5 |
| YOLOv11 | 60.1 | 44.5 | 46.0 | **65.5** | **65.7** | **65.1** | 65.4 | 65.4 | 21.6 | 6.3 | 1.6 | 66.9 | 65.3 | **72.8** | **68.0** |
| RetinaNet | 57.5 | 47.2 | 43.8 | **62.3** | **66.3** | **60.3** | 63.4 | 66.3 | 43.8 | 14.6 | 4.7 | **63.4** | 66.0 | **73.2** | 63.1 |
| Faster R-CNN | 58.0 | 47.0 | 42.2 | **63.1** | **66.1** | **61.0** | 65.7 | 66.9 | 34.6 | 18.7 | 11.7 | 61.5 | 65.9 | **71.6** | 64.0 |
| Deformable R-CNN | 59.4 | 48.2 | 45.0 | **65.2** | **66.3** | **65.4** | 66.4 | 68.7 | 34.1 | 15.7 | 6.7 | 63.3 | **66.8** | **72.3** | **66.3** |
| SSD | 39.3 | 22.3 | 22.8 | **46.1** | **43.7** | **45.1** | **44.2** | 49.0 | 21.9 | 5.2 | 1.9 | 50.1 | 42.5 | **55.2** | 50.6 |
| DETR | 43.2 | 29.4 | 26.2 | **54.1** | **54.3** | **48.4** | 31.9 | 41.9 | 22.0 | 8.4 | 3.3 | 54.8 | **53.2** | **63.8** | **57.1** |
| DINO | 59.9 | 50.3 | 46.3 | **64.2** | **67.6** | **64.1** | 51.3 | 48.9 | 32.0 | 17.6 | 7.1 | 57.0 | **68.0** | **74.9** | 64.6 |
| CZ Det | 50.7 | 30.6 | 33.8 | **67.5** | **62.5** | **66.8** | 56.9 | 52.5 | 13.0 | 7.3 | 0.2 | 61.8 | 60.5 | **69.7** | **72.6** |

**Effect of pose diversity on generalizability.** In SAR tasks, it is important to collect data of individuals in a variety of poses since missing persons in forest environments may be found in diverse postures. However, most existing public person detection datasets predominantly consist of upright individuals, with standing poses comprising the vast majority. We hypothesize that this imbalance limits the generalizability of person detection models for SAR applications. To validate this hypothesis, we conduct an experiment using ForestPersons with pose annotations, where we intentionally construct standing-dominant training set to mimic the pose distribution of existing datasets. Specifically, we trained object detection models using only samples labeled with standing poses and evaluated their performance on test samples categorized into standing, sitting, and lying poses, respectively.

The results are presented in the Table 4(a). Specifically, models trained solely on standing attribute exhibited significantly lower performance in detecting sitting and lying poses across all evaluated models. In contrast, models trained on the dataset with comprehensive pose annotations, achieved improved detection performance across all pose categories. These findings highlight the importance of collecting diverse human poses for SAR tasks. ForestPersons addresses this need by including underrepresented poses such as sitting and lying, which are often absent from conventional public datasets, making it more suitable for under-canopy person detection in SAR scenarios.

**Effect of season diversity on generalizability.** The visual appearance of forest environments can vary across seasons due to changes in under-canopy vegetation density, foliage, and lighting conditions. These seasonal differences directly affect the visibility and occlusion patterns of individuals, which in turn influence detection difficulty. We assume that insufficient seasonal diversity in training data constrains the generalization capability of detection models under diverse environmental conditions. To demonstrate this, we conduct a controlled experiment using ForestPersons with explicit season labels, comparing models trained on a specific season and tested on different seasons.

The results on the Table 4(b) show a clear asymmetry in cross-season performance. Models trained on only summer images exhibited performance degradation when tested on winter images but maintained a relatively stable level of AP. In contrast, models trained solely on winter images showed a significant drop in performance when evaluated on summer and fall images. Notably, when models were trained on images from all seasons, they achieved consistent performance across all seasonal conditions. These findings highlight the importance of seasonally diverse training data for SAR mission success, which our dataset fulfills by including images captured across different seasons.

Table 5: Model performance comparison between the Faster-RCNN model trained with motion-blur augmented data, and the model trained with original data with respect to real MAV dataset.

| Train dataset | AP | $AP_{50}$ | $AP_{75}$ | AR |
|---|---|---|---|---|
| ForestPersons (artifact-free) | **61.4** | **88.4** | **76.6** | **69.0** |
| ForestPersons (augmented) | 35.8 | 64.7 | 33.9 | 49.5 |
| SARD | 23.2 | 53.5 | 18.5 | 34.8 |
| HERIDAL | 0.0 | 0.0 | 0.0 | 0.3 |
| WiSARD | 40.2 | 75.2 | 35.0 | 50.0 |

## 4.5 EVALUATING SAR-TRAINED MODELS ON A REAL MAV DATASET

ForestPersons was collected in a controlled setting using handheld and tripod-mounted cameras rather than an aerial platform, which may introduce a domain gap from real MAV footage due to drone-specific artifacts such as motion blur and sensor noise. To verify whether such a gap exists and whether augmenting training data with simulated artifacts can bridge it, we conducted additional experiments by collecting a new test dataset of 24,209 images using an actual MAV, as shown in Fig. 7(a) in Appendix. This new dataset therefore reflects real MAV imaging conditions, including platform-induced degradations such as occasional motion blur and sensor noise. We recorded two individuals in standing, sitting, and lying postures across multiple background settings.

**Training on ForestPersons vs. artifact-augmented ForestPersons.** To quantitatively assess the potential domain gap, we created an *augmented version* of ForestPersons by applying motion blur and ISO noise using the `Albumentations` library (Buslaev et al., 2020) to simulate MAV-specific artifacts. Specifically, we apply motion blur with a blur limit of [3, 30] and ISO noise with color shift and intensity ranges of [0.01, 0.05] and [0.1, 0.3], respectively. All augmentations are applied with probability 1.0, assuming these artifacts are consistently present in MAV-captured imagery. Visual examples of these augmented images are provided in Fig. 22 in Appendix.

Subsequently, we evaluated two Faster R-CNN models on the real-world MAV test set: one trained on the original ForestPersons and the other on its augmented version. As shown in Table 5, the artifact-free model achieved 61.4 AP, while the augmented model dropped to 35.8 AP. This demonstrates that our handheld collection method generalizes well to the actual MAV domain. Our qualitative analysis of the MAV dataset supports this finding, as the vast majority of frames are artifact-free, with significant motion artifacts appearing only infrequently during occasional abrupt maneuvers.

**Training on prior SAR datasets.** We additionally benchmark models trained on prior SAR datasets on the same real-world MAV test set. As shown in Table 5, they achieved lower AP due to the domain gap between their high-altitude viewpoints and our low-altitude, under-canopy setting, despite being captured from actual drones. Qualitative predictions for all models are shown in Fig. 23 in Appendix.

Although ForestPersons was not collected using an actual MAV, the comparative results on the real-world MAV test set indicate that models trained on ForestPersons achieve higher performance than those trained on artifact-augmented ForestPersons or existing drone-captured SAR datasets. These findings suggest that ForestPersons provides an effective training source for real MAV deployment, despite its handheld data collection setting.

## 5 DISCUSSION AND CONCLUSION

ForestPersons is the first large-scale dataset designed to detect missing persons in under-canopy forest environments. Unlike previous SAR benchmarks that focus on UAV-based aerial imagery, ForestPersons provides ground-level views from the perspective of MAVs, which are more suitable for detecting partially occluded individuals beneath forest canopies. The dataset includes annotations for various attributes, such as season, location type, weather, human pose, and visibility level, providing a basis for training and evaluating models under diverse and realistic SAR scenarios. By addressing the critical bottleneck of data scarcity in under-canopy SAR scenarios, ForestPersons formalizes missing-person detection in such environments as a benchmarkable and reproducible research problem, enabling systematic development and evaluation of AI models for autonomous SAR.

**Ethics Statement.**  All scenes in ForestPersons consist of staged missing person scenarios with voluntary participants, ensuring safety and ethical compliance. No real missing person cases are included. Face anonymization was applied as described in Section 3.2, ensuring that no personal or identifiable information remains. The dataset will be released under a research-only license, and responsible and transparent use is strongly encouraged any harmful or military use is strictly prohibited.

**ACKNOWLEDGMENTS**  This work was supported by Institute of Information & communications Technology Planning & Evaluation (IITP) grant funded by the Korea government(MSIT) (No.2022-0-00021, Development of Core Technologies for Autonomous Searching Drones)

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

## A   MISSING PERSON DETECTION IN AUTONOMOUS SAR SYSTEM

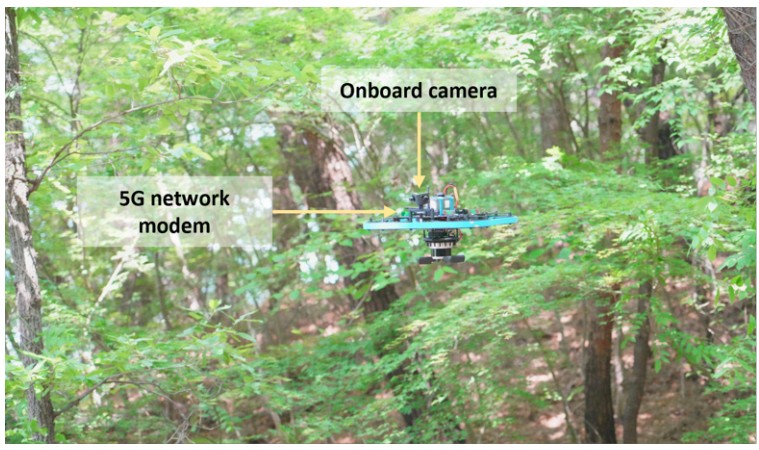
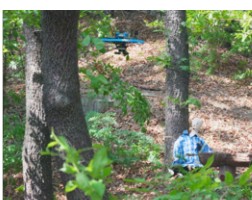
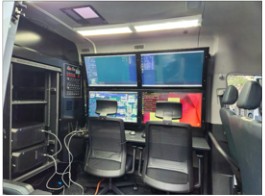

(a) Autonomous flight MAV

(b) Onboard Inference

(c) Edge Server Inference

Figure 7: **Missing person detection inference in autonomous SAR systems** (a) Autonomous Flight: A MAV performs a search mission under the forest canopy. (b) Onboard Inference: Frames captured by the onboard camera are processed locally on the MAV in real time using a lightweight detection model. (c) Edge Server Inference: Frames are transmitted via a commercial 5G network to a remote edge server, where inference is performed using a higher-capacity model.

As a future direction and an ongoing application of ForestPersons, we configured a missing person detection pipeline for autonomous SAR missions. In this setup, frames captured by the onboard camera of a Micro Aerial Vehicle (MAV) flying under forest canopy conditions are processed by detection models trained on ForestPersons, which are deployed either onboard the MAV or on a remote edge server depending on mission requirements.

These two inference paths are selected based on the trade-off between latency, bandwidth, and model complexity. For onboard inference, lightweight object detection models such as variants of YOLO (Redmon et al., 2016) are optimized using tools such as NVIDIA TensorRT or Intel OpenVINO to meet real-time constraints on resource-limited hardware, which is particularly useful when low-latency response and independence from network connectivity are critical. In contrast, edge inference allows the use of more advanced models such as transformer-based state-of-the-art architectures like DINO (Caron et al., 2021). In this case, video streams are transmitted over a high-bandwidth wireless communication system, such as 5G, to a remote server with greater computational resources. This enables the use of advanced detection algorithms that leverage greater computational resources to achieve improved performance compared to what is feasible on resource-constrained onboard systems.

Figure 7 illustrates this architecture: (a) The MAV autonomously performs a low-altitude search mission under the forest canopy. (b) Each captured frame is processed in real time onboard using an optimized lightweight detection model. (c) Alternatively, the video stream is transmitted over a commercial 5G network to a remote edge server, where inference is performed by a more powerful model.

Field experiments were conducted under canopy conditions using both mannequins and human actors to simulate missing persons. In these trials, both onboard and edge inference modes successfully detected targets in realistic environments, demonstrating the effectiveness of our SAR system and the applicability of ForestPersons to real-world scenarios. Specifically, to evaluate the domain gap between ForestPersons and real-world MAV data, we curated the real MAV Flight dataset; the corresponding experiments are reported in Section 4.5.

# B BENCHMARK MODELS

## B.1 IMPLEMENTATION DETAILS

Table 6: **Hyperparameter settings for training object detections.** Most configurations follow the default setting of MMDetection and detrex.

| Methods | Optimizer | Learning rate | Batch size | Weight decay | Epoch |
|---|---|---|---|---|---|
| YOLOv3 (Redmon & Farhadi, 2018) | SGD | $1 \times 10^{-3}$ | 64 | $5 \times 10^{-4}$ | 273 |
| YOLOX (Ge et al., 2021) | SGD | $1 \times 10^{-2}$ | 64 | $5 \times 10^{-4}$ | 300 |
| YOLOv11 (Jocher & Qiu, 2024) | SGD | $1 \times 10^{-2}$ | 16 | $5 \times 10^{-4}$ | 100 |
| RetinaNet (Lin et al., 2017) | SGD | $5 \times 10^{-3}$ | 16 | $1 \times 10^{-4}$ | 12 |
| Faster R-CNN (Ren et al., 2015) | SGD | $2 \times 10^{-2}$ | 16 | $1 \times 10^{-4}$ | 12 |
| Deformable Faster R-CNN (Jocher & Qiu, 2024) | SGD | $2 \times 10^{-2}$ | 16 | $1 \times 10^{-4}$ | 12 |
| SSD (Liu et al., 2016) | SGD | $1.5 \times 10^{-2}$ | 192 | $4 \times 10^{-5}$ | 120 |
| DETR (Carion et al., 2020) | AdamW | $1 \times 10^{-4}$ | 16 | $1 \times 10^{-4}$ | 150 |
| DINO (Caron et al., 2021) | AdamW | $1 \times 10^{-4}$ | 16 | $1 \times 10^{-4}$ | 12 |
| CZ Det (Meethal et al., 2023) | SGD | $1 \times 10^{-2}$ | 32 | $1 \times 10^{-4}$ | 30 |

In this section, we describe the hyperparameter settings used to train each object detection model for benchmarking purposes. Table 6 summarizes the configurations for all models. Most hyperparameters follow the default settings provided by the MMDetection (Chen et al., 2019), detrex (Ren et al., 2023) and detectron2 (Wu et al., 2019) frameworks, except RetinaNet, for which we reduced the learning rate compared to the default setting to prevent training instability observed with higher values.

## B.2 ANALYSIS OF BENCHMARK MODELS

Given the critical nature of SAR missions, achieving high recall is a primary requirement, necessitating a more deliberate examination of the precision-recall trade-off compared to conventional object detection tasks. To investigate this aspect, we present the precision-recall curve at an IoU threshold of 0.5, as illustrated in Figure 8. The precision-recall curve is constructed by sorting predicted bounding boxes in descending order of confidence scores , with increasing confidence thresholds prioritizing precision over recall, while lower thresholds capture more true positives at the cost of introducing false positives. The resulting shape of the curve characterizes how each model behaves under varying confidence thresholds, offering insight into its sensitivity to recall-focused operating points.

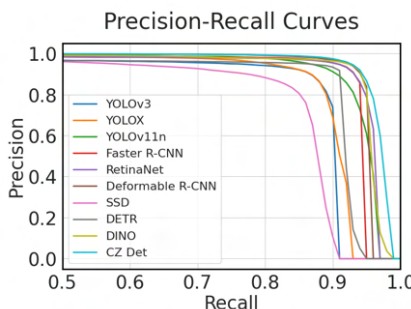

Figure 8: **Precision-Recall curves of baseline object detection models.**

In Figure 8, CZ Det (Meethal et al., 2023) (cyan) consistently maintains high recall even at low confidence thresholds, whereas SSD (Liu et al., 2016) (pink) exhibits a clear limitation in its recall capacity. Specifically, even when all predicted bounding boxes are treated as true positives, its curve saturates below the recall levels reached by DINO. This indicates a structural limitation in SSD's detection capability that cannot be overcome by threshold tuning alone. Such findings indicate that, particularly in SAR contexts, the upper bound of recall achievable by a model constitutes an essential metric in itself, complementing traditional aggregate measures such as mAP.

In practice, the confidence threshold is often selected based on the point that maximizes the F1-score, calculated on a validation or a test set. However, in recall-sensitive domains such as SAR, it may be more appropriate to deliberately reduce the threshold to prioritize recall, even at the expense of an increased false positive rate. This strategy aligns with real-world operational considerations, wherein human operators may prefer investigating more detection candidates rather than risking failure to detect actual missing persons. Therefore, we argue that the development and evaluation of object detectors for SAR applications should incorporate not only AP but also (1) the maximum attainable recall and (2) the recall level at which precision begins to decline sharply. These indicators are closely

tied to the likelihood of successfully locating and rescuing missing persons, and thus serve as critical performance criteria in SAR applications.

## C    CASE STUDY: SUCCESSES, FAILURES, AND FUTURE DIRECTIONS

### C.1    LIMITATIONS OF GENERALIZATION FROM PRIOR BENCHMARKS

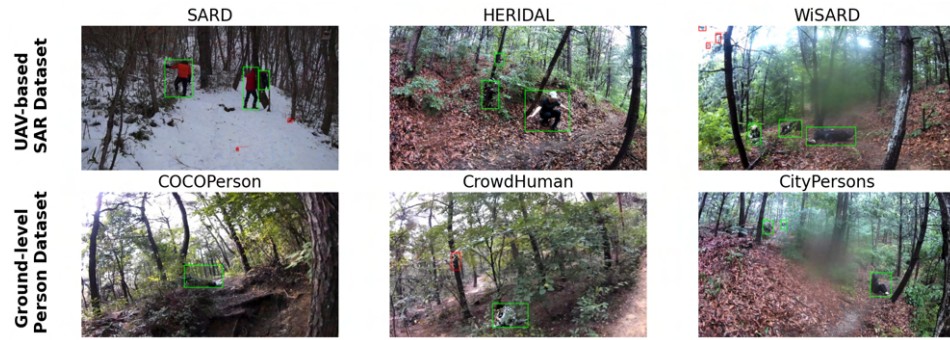

Figure 9: **Failure cases of object detection models trained on prior UAV-based SAR and Ground-level Person datasets.** Green boxes indicate ground-truth bounding boxes, and red boxes represent model predictions. These examples illustrate the limitations of existing datasets in handling under-canopy SAR scenarios.

ForestPersons differs from conventional detection benchmarks in several key aspects, including viewpoint, environmental complexity, and the conditions of human targets. To assess generalizability, we evaluated models trained on existing SAR datasets and ground-level person datasets using the ForestPersons test split. Specifically, we selected SARD (Sambolek & Ivasic-Kos, 2021), HERIDAL (Kundid Vasić & Papić, 2022), and WiSARD (Broyles et al., 2022) as representative UAV-based SAR datasets, and COCOPersons Lin et al. (2014), CrowdHuman (Shao et al., 2018), and CityPersons (Zhang et al., 2017) as representative ground-level person datasets. For all experiments, we used Faster R-CNN (Ren et al., 2015) as the object detection model.

As illustrated in Figure 9, models trained on these existing datasets exhibit limited generalizability when applied to under-canopy SAR scenarios. This outcome is expected: prior UAV-based SAR datasets primarily contain aerial images, which differ significantly from the ground-level perspectives that are characteristic of under-canopy tasks. While ground-level datasets more closely reflect the viewpoint of MAV flights compared to conventional UAV-based SAR datasets, they still predominantly feature upright and fully visible individuals. Consequently, they fall short in representing challenging cases such as non-standing or heavily occluded persons, which are common in forest search scenarios.

### C.2    EVALUATION ON FORESTPERSONS

We then evaluated a model trained on the ForestPersons training split to assess the detection performance gains from using data specifically designed to reflect under-canopy SAR conditions. As shown in Figure 10, the Faster R-CNN model trained on ForestPersons successfully detects missing persons that were not captured by models trained on prior UAV-based or ground-level dataset. This provides qualitative evidence that our dataset better suits SAR tasks in under-canopy environments.

We further investigated the factors contributing to prediction failures on the ForestPersons test set, even when using models trained on ForestPersons. Specifically, we analyzed the prediction results of a Faster R-CNN model trained on ForestPersons by visualizing the confusion matrix, as

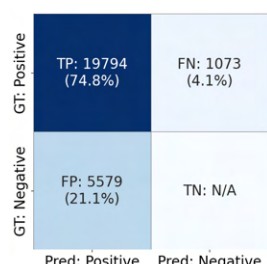

Figure 11: **Confusion matrix of the object detection model trained with ForestPersons.**

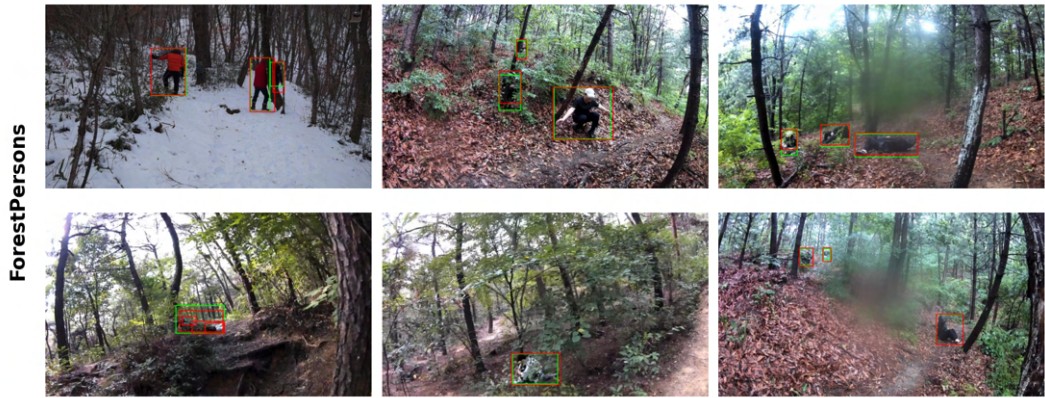

Figure 10: **Success cases of object detection models trained on ForestPersons.** Green boxes indicate ground-truth bounding boxes, and red boxes represent model predictions. The models trained with ForestPersons detect the failure case of the models trained with the existing dataset.

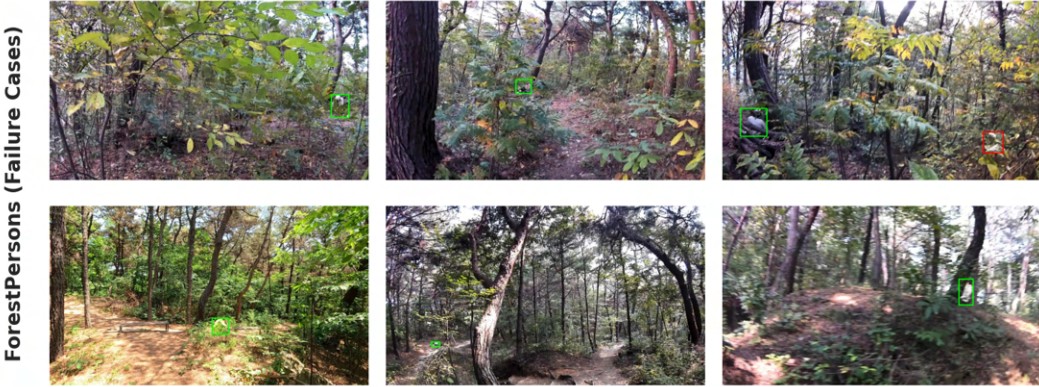

Figure 12: **Failure cases of object detection models trained on ForestPersons.** Green boxes indicate ground-truth bounding boxes, and red boxes represent model predictions. Ground-truth instances with high occlusion or small bounding box size tend to be frequently missed by the detection model.

shown in Figure 11. The confusion matrix summarizes all predictions on the test set and reveals that false positives significantly outnumber false negatives.

However, given the critical nature of SAR tasks, where false negatives are significantly more detrimental than false positives, we focused our analysis on ground-truth instances that were classified as false negatives. Figure 12 presents visual examples of these cases. As expected, the model struggled to detect individuals with small bounding boxes or under heavy occlusion by natural obstacles.

Interestingly, the winter subset yields noticeably fewer false negatives, suggesting that winter images are generally less challenging for the detection model. A plausible explanation is that individuals in winter scenes are more visually salient due to the higher contrast between individuals and the snow-covered background, which facilitates easier detection. This explanation is supported by the experiment in Table 4(b) where a model trained exclusively on winter images generalized worse to the test set than a model trained only on summer images. This indicates that winter images may lack sufficient variability to support effective generalization, which is why they are easier for missing person detection, ultimately reducing the likelihood of false negatives. In contrast, summer images, which often contain dense vegetation leading to various occlusions, contribute more to the generalization ability of the model. These qualitative and quantitative findings help us understand the exceptionally low incidence of false negatives in winter images.

## C.3 Generalization Failures from Limited Attribute Training

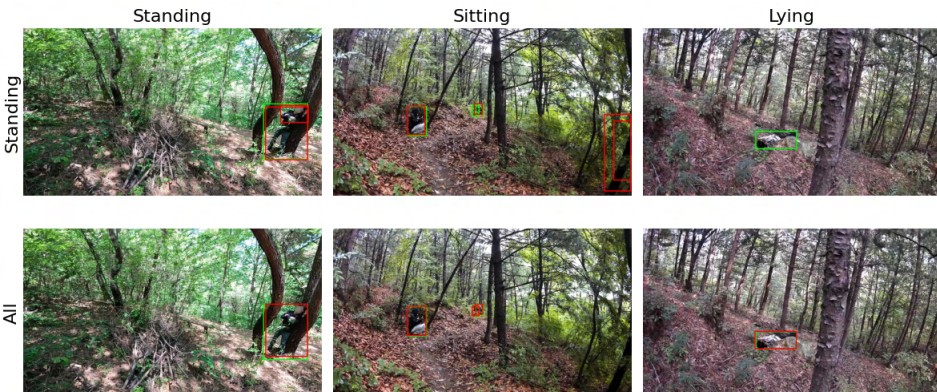

Figure 13: **Detection cases for models trained on standing-only data and all-pose data.** Green boxes indicate ground-truth bounding boxes, and red boxes represent model predictions. Each row corresponds to a model, and each column corresponds to the ground truth pose in the test image. Models trained on a specific pose often fail to detect individuals in other poses and sometimes identify incorrect regions as humans.

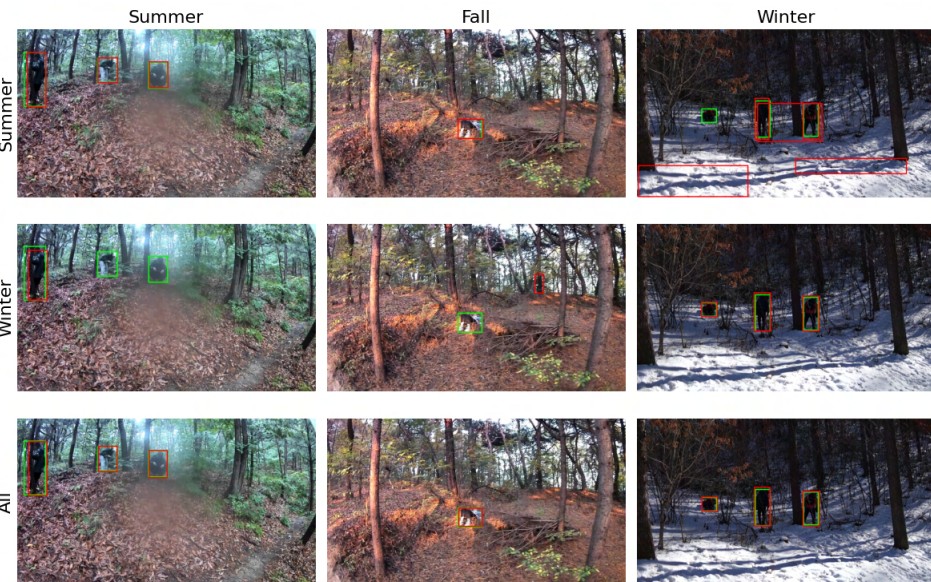

Figure 14: **Detection cases for models trained on summer-only, winter-only, and all-season data.** Green boxes indicate ground-truth bounding boxes, and red boxes represent model predictions. Each row shows detection from a model trained on data from a specific season, while each column represents test data from a particular season. Models trained on limited seasonal data show clear seasonal bias when applied to scenes from different seasons, such as failing to detect people or generating inaccurate bounding boxes.

Extending the results shown in Table 4, we further analyzed how restricting training data to specific attributes, such as pose or season, affects the performance of Faster R-CNN. In Figure 13, models trained only on standing poses perform poorly when detecting people in other postures, such as sitting or lying. These models mainly respond to upright shapes, often mistaking vertical objects like tree trunks for people, and failing to detect people who are lying on the ground. This indicates that the model has become overly reliant on shape cues associated with upright postures observed during training, and consequently fails to generalize to sitting or lying poses.

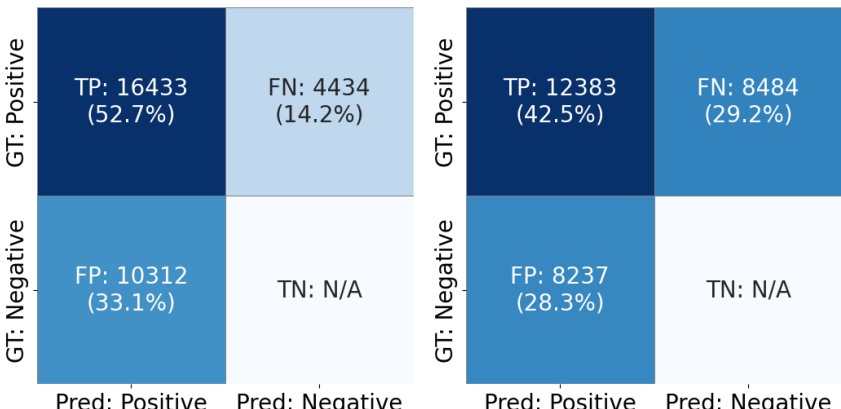

Figure 15: **Confusion matrices of object detection models trained on summer and winter datasets.** (Left) Summer-trained model; (Right) Winter-trained model.

A similar pattern is seen in the seasonal experiments in Figure 14. Models trained only on summer images, which contain more vegetation and frequent occlusion, show slightly better generalization to other seasons. The presence of dense vegetation and natural occlusion in summer scenes appears to help the model learn features that generalize better to different seasonal environments. However, these models still produce many errors in winter scenes, such as false positives caused by mistaking snow-covered terrain for people. In contrast, models trained only on winter images perform significantly worse in other seasons. Winter scenes usually lack vegetation and have fewer occluding elements, which limits the diversity of visual cues the model can learn from. As a result, these models often fail to detect people in summer scenes with dense foliage and complex backgrounds, leading to frequent false negatives. This tendency is reflected in the confusion matrices shown in Figure 15. These findings indicate that the visual properties of each season shape how the model learns and where it tends to fail, and that training on a single season is not sufficient to ensure robustness across seasonal conditions.

Unlike models trained on a single season or pose, the model trained on the complete dataset, which includes a full range of poses and seasonal conditions, performs more reliably, as shown in the last rows of Figure 13 and Figure 14. These results demonstrate the effectiveness of ForestPersons as a benchmark that reflects the diversity and complexity of real-world SAR conditions. By providing extensive variation in human pose, occlusion, and environmental factors, ForestPersons supports the development of more generalizable models and serves as a solid foundation for advancing robust missing person detection in challenging under-canopy search tasks.

### C.4 LIMITATIONS EXPOSED AND DIRECTIONS FOR FUTURE SAR DETECTION

Our qualitative analysis highlights the utility of ForestPersons in diagnosing the generalization and structural limitations of representative detection models in the context of SAR missions. ForestPersons introduces new challenges by incorporating vegetation-rich environments that frequently cause occlusion, diverse human poses including non-upright postures, and seasonal conditions such as snow that are often absent in prior datasets. These findings show that models trained on narrow visual patterns may seem reliable in simplified test environments but fail to maintain the same level of reliability when applied to real-world conditions. While ForestPersons was carefully designed to cover a wide range of poses, occlusion levels, and seasonal conditions, our analysis suggests that some failure cases may still remain undetected. Dataset diversity is therefore critical for revealing model limitations, but it alone may not be sufficient.

To address this, complementary approaches such as optimizing viewpoint and trajectory design can further reduce the inherent difficulty of the detection task and enhance practical performance in the field. One such approach is viewpoint-aware flight planning, which can support vision models by improving the visibility of missing persons. By explicitly accounting for the MAV's camera field of view, such planning can help ensure that individuals are captured from favorable angles and

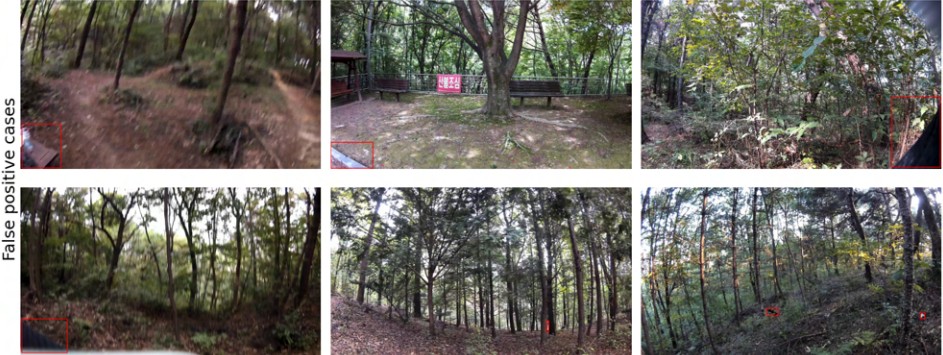

Figure 16: **Examples of false positive cases in person-absent situations.** Red boxes represent model predictions.

distances. In contrast, coarse trajectories that simply follow major roads may expose the model to less informative and more occluded perspectives. Therefore, alongside the use of diverse datasets like ForestPersons, flight strategies that structurally facilitate detection should be explored as a complementary direction, particularly in the context of autonomous SAR missions.

### C.5   EVALUATION ON PERSON-ABSENT SITUATIONS

ForestPersons was primarily designed to capture realistic SAR scenarios in which a missing person is present. Such data is significantly more difficult and costly to collect and annotate than person-absent forest imagery, which is comparatively easier to obtain. Our initial focus was therefore on ensuring high-quality coverage of person-present situations.

To examine model behavior in person-absent settings, we curated a separate set of 193 images without humans and evaluated a model trained solely on ForestPersons. The model produced 13 false detections, corresponding to a false positive rate (FPR) of approximately 6.7%. This negative set is publicly released in a separate directory, 379_FPV_No_Person_summer_forest, so that researchers can directly benchmark false positive performance under person-absent conditions. Representative false positive cases are shown in Fig. 16.

Looking forward, we plan to extend ForestPersons by systematically including additional person-absent imagery using our MAV-based collection system (Fig. 7). This expansion will provide more balanced coverage of positive and negative cases, enabling comprehensive training and benchmarking of models under realistic SAR conditions.

## D   DATA COLLECTION GUIDELINES

ForestPersons was constructed to reflect realistic search scenarios for missing persons in forested environments. All video sequences were recorded using handheld or tripod-mounted cameras, including GoPro HERO 9 Black, Sony SLT-A57, and See3CAM 24CUG models. The cameras were positioned to simulate the typical flight altitudes and viewing angles of low-altitude MAVs operating under forest canopy, capturing slightly downward-facing perspectives similar to those used in actual search operations. All recordings were captured at a frame rate of at least 20 FPS, with resolution settings adjusted depending on the camera model used.

### D.1   LOCATIONS: FOREST ENVIRONMENTS RELEVANT TO SAR MISSIONS

All data were collected in forested regions where real-world missing person incidents are likely to occur. We selected diverse environments including dense forest interiors, valleys, and forest entrances to reflect typical terrain encountered during SAR missions. These locations span a range of vegetation density and visibility conditions, from heavily occluded forest interiors to forest edge regions with sparse vegetation.

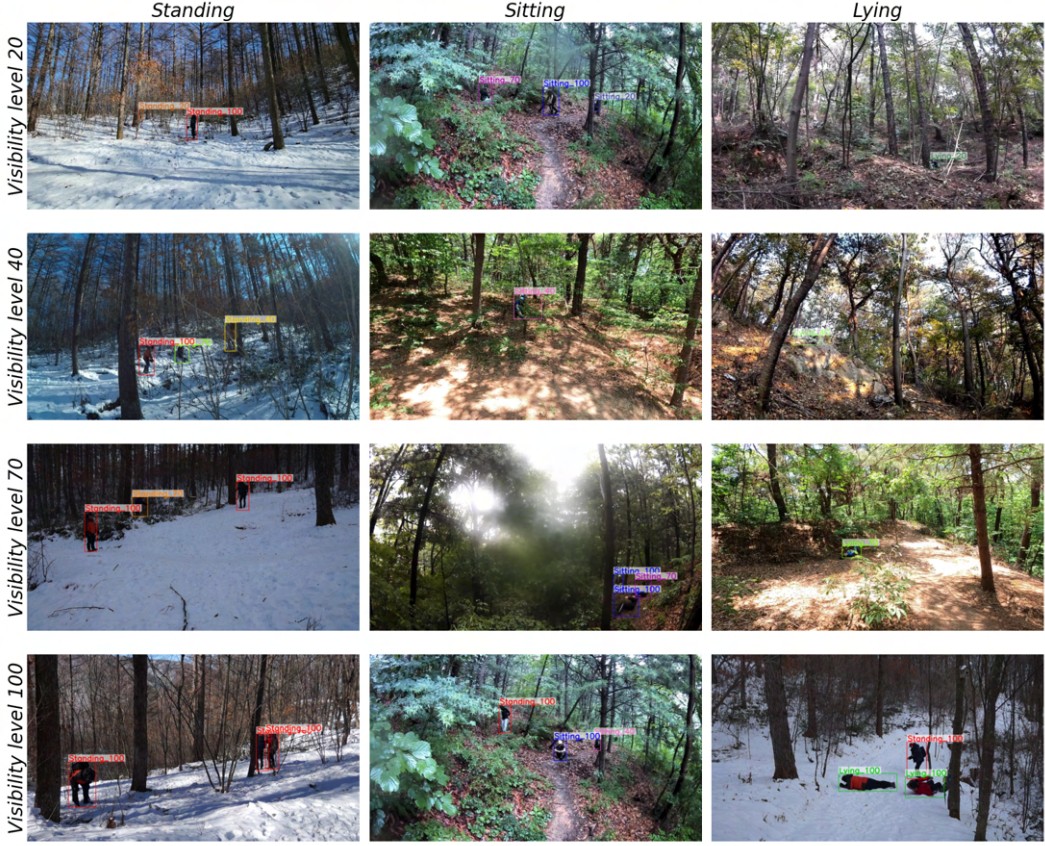

Figure 17: **Examples across various visibility levels and poses.** Images are grouped by visibility level (rows) and pose (columns), each drawn from distinct scene contexts.

Each environment includes natural sources of visual occlusion such as tree branches, underbrush, uneven terrain, and varying vegetation density. We aimed to incorporate diverse spatial layouts that challenge missing person detection, including not only typical forest trails but also rocky valleys and steep slopes covered with dense foliage. This diversity enables the dataset to capture a broad range of search scenarios encountered in SAR missions.

## D.2 WEATHER AND TIME OF DAY

To reflect the environmental diversity encountered in real-world search operations, data were collected under various weather and lighting conditions. All video sequences were captured during daytime or twilight hours before sunset, when there was sufficient natural light. Night time scenes were excluded due to safety concerns during field deployment and the limited effectiveness of RGB-based detection in low-light conditions.

Weather and seasonal conditions included sunny, overcast, and snow-covered winter environments. These variations allowed us to capture diverse visual appearances, including strong shadows under direct sunlight, diffuse lighting on cloudy days, and high reflectance and severe occlusion in snowy terrain. Each sequence is accompanied by metadata describing both the season and weather, enabling evaluations under specific environmental contexts.

## D.3 SUBJECT BEHAVIOR AND CAPTURE STRATEGY

To simulate realistic SAR scenarios, actors in the ForestPersons performed a wide range of behaviors, including standing, sitting, lying down, and natural transitions between these states. Transitional poses (e.g., moving from a seated to a standing position) were annotated with the nearest posture

label, typically sitting or standing. Although this labeling may involve some degree of annotator subjectivity, its impact on the overall data quality is minimal. These behavioral variations reflect the diversity of human configurations encountered in SAR operations.

Camera platforms included handheld rigs and tripods. To emulate the viewpoint of MAVs operating under canopy, operators followed movement paths consistent with low-altitude MAV trajectories. Camera height, angle, and distance were varied within and across sequences to simulate oblique and horizontal viewpoints. This variation allowed us to capture human subjects from perspectives representative of realistic aerial search conditions.

A key aspect of our strategy was the active creation of natural occlusion. Rather than using fixed occlusion setups, camera operators navigated around tree branches, bushes, or through dense vegetation to partially obscure subjects in dynamic and realistic ways. In difficult environments such as snowy or rainy terrain, where operator movement posed safety risks, the camera was fixed and actors moved within the frame to simulate occlusion safely.

## E    VIDEO SEQUENCE-LEVEL DIFFICULTY ESTIMATION

ForestPersons was collected as a set of video sequences, from which image frames were extracted to construct the final dataset. In this setup, if frames from the same sequence are split across training, validation, and test splits, it can lead to overestimated model performance. This is because detection models may implicitly learn scene-specific backgrounds or appearances during training, and then encounter similar contexts during evaluation, resulting in inflated accuracy that does not reflect true generalization. To avoid such overlap, we split the dataset at the sequence level, ensuring that each video sequence appears in only one of the train, validation, or test splits.

### E.1    NECESSITY OF DIFFICULTY-AWARE DATA SPLITTING

A naive approach such as randomly assigning sequences to each split, or manually selecting them based on subjective judgment (e.g., "easy-looking" or "challenging" scenes), can lead to distributional bias across splits. For example, one split might inadvertently contain mostly clear and well-lit scenarios, while another might be dominated by occluded or low-visibility scenes. Such imbalance can undermine the fairness and interpretability of model comparisons.

To mitigate this issue, we introduced a model-based method for estimating sequence-level difficulty, providing a principled way to assess and distribute difficulty across the dataset.

### E.2    MODEL-BASED DIFFICULTY ESTIMATION

We employed a Faster R-CNN (Ren et al., 2015) object detector pretrained on the COCO (Lin et al., 2014) dataset to estimate the detection difficulty of each sequence. For each sequence, we applied the detector to all images and computed the Average Precision (AP). The difficulty score for a sequence $s$ is then defined as:

$$\text{Difficulty}(s) = 1 - \text{AP}_{50}(s) \tag{1}$$

Here, $\text{AP}_{50}(s)$ denotes the performance of the detector model on sequence $s$, averaged over all annotated frames. Higher AP values indicate that the sequence is easier to detect, while a lower AP corresponds to more challenging scenes. This formulation provides an objective difficulty measure, independent of annotator intuition or handcrafted heuristics.

### E.3    DIFFICULTY-AWARE DATASET SPLITTING

Based on the estimated difficulty scores, we sorted all video sequences in ascending order of AP (i.e., increasing difficulty) and allocated them to train, validation, and test splits to ensure balanced difficulty distribution. For example, sequences were interleaved across splits so that each contained a diverse mixture of easy, medium, and hard samples.

As shown in Figure 18, the difficulty curve of ForestPersons illustrates that each sequence spans a range of detection difficulty. Each point corresponds to a video sequence, sorted by its model-based

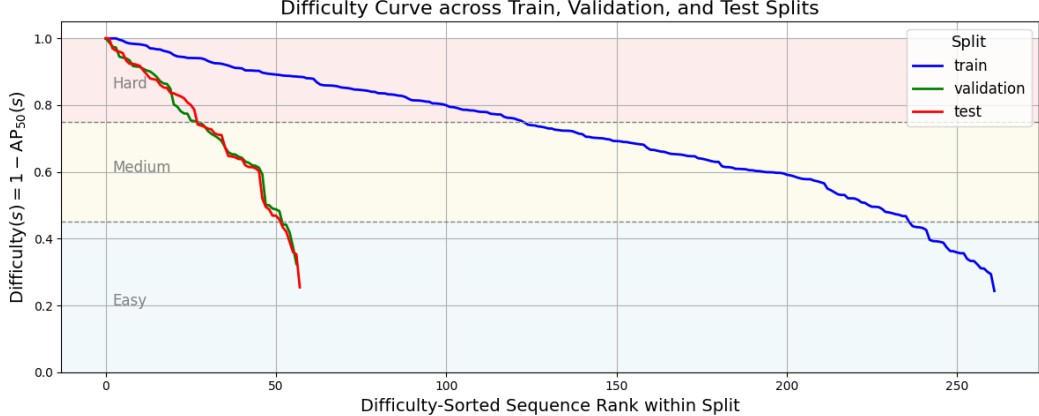

Figure 18: Difficulty Curve across Train, Validation, and Test Splits

difficulty score $1 - \mathrm{AP}_{50}(s)$. The plot illustrates that the dataset spans a broad range of difficulty levels, ensuring balanced evaluation across splits.

## F    QUANTITATIVE ANALYSIS OF ANNOTATION QUALITY

To evaluate the consistency and reliability of our annotations, we conducted a controlled user study with six independent annotators who were not involved in the original labeling process. Each annotator was provided with our annotation guideline (Section 3.2) but had no access to the original labels. We randomly sampled 241 images, covering 12 combinations of pose and visibility attributes and including a total of 525 bounding boxes.

Table 7 reports performance metrics with respect to the ground-truth boxes, where a user annotation was considered correct if the IoU exceeded 0.5. Annotators generally achieved high precision, recall, and F1 scores, indicating reliable bounding box quality.

Table 7: Bounding Box Inter-Annotator Agreements.

|  | Annotator A | Annotator B | Annotator C | Annotator D | Annotator E | Annotator F |
|---|---|---|---|---|---|---|
| mean IoU | 0.8112 | 0.7936 | 0.8249 | 0.8047 | 0.7749 | 0.7975 |
| Precision | 0.9432 | 0.9226 | 0.9298 | 0.9374 | 0.8643 | 0.9366 |
| Recall | 0.9181 | 0.9086 | 0.9333 | 0.8552 | 0.8495 | 0.8438 |
| F1 Score | 0.9305 | 0.9155 | 0.9316 | 0.8944 | 0.8569 | 0.8878 |
| True Positive | 482 | 477 | 490 | 449 | 446 | 443 |
| False Positive | 29 | 40 | 40 | 37 | 70 | 30 |
| False Negative | 43 | 48 | 35 | 76 | 79 | 82 |

Table 8 reports the overall inter-annotator agreements for pose and visibility attributes, while Table 9 presents Cohen's $\kappa$ values across annotators for each attribute. Pose labels achieved high agreement (Percent Agreement $\approx 0.89$, Cohen's $\kappa > 0.83$ Cohen (1960), and Fleiss' $\kappa = 0.7414$ Fleiss (1971)), indicating that annotators could reliably distinguish between different human poses. By conventional interpretation of Cohen's $\kappa$, values above 0.81 are regarded as *almost perfect* agreement. In contrast, visibility attributes showed relatively lower agreement (Percent Agreement $\approx 0.62$, Cohen's $\kappa \approx 0.45$, Fleiss' $\kappa = 0.5048$), corresponding to the *moderate* agreement range (0.41–0.60).

We note that these challenges are not unique to ForestPersons. The difficulty of consistently labeling occlusion has been widely reported across benchmarks involving partially visible humans. For example, in the CrowdHuman dataset (Shao et al., 2018), annotators are instructed to complete the full-body bounding box even when the person is partially hidden, which often introduces variance due to differing subjective interpretations. Similarly, COCO-OLAC (Wei et al., 2025) defines occlusion levels using estimated occlusion ratios, requiring annotators to mentally reconstruct invisible body

Table 8: Attributes Inter-Annotator Agreements.

|  | Annotator A | Annotator B | Annotator C | Annotator D | Annotator E | Annotator F |
|---|---|---|---|---|---|---|
| Pose | 0.8963 | 0.8952 | 0.8577 | 0.8976 | 0.8677 | 0.9142 |
| Visibility levels | 0.6183 | 0.6310 | 0.6163 | 0.6192 | 0.6188 | 0.6027 |

Table 9: Attributes Cohen's $\kappa$ across annotators.

|  | Annotator A | Annotator B | Annotator C | Annotator D | Annotator E | Annotator F |
|---|---|---|---|---|---|---|
| Pose | 0.8396 | 0.8346 | 0.8577 | 0.8410 | 0.7952 | 0.8654 |
| Visibility levels | 0.4514 | 0.4588 | 0.4574 | 0.4537 | 0.4457 | 0.4271 |

parts from context. The HOOT (Sahin & Itti, 2023) explicitly draws occlusion masks and categorizes occlusion types, while OVIS (Qi et al., 2022) adopts bounding-box occlusion rates (BOR) derived from overlaps. Even in OVIS, where IoU-based measures provide more objective criteria, some degree of subjectivity remains unavoidable in the initial classification process. In summary, although recent benchmarks attempt to quantify occlusion with numerical ratios or overlap measures, the process of determining visibility levels cannot be fully disentangled from heuristic estimation.

Collectively, these cases demonstrate that the subjectivity and ambiguity we encountered in visibility labeling are not exceptions. Rather, they represent intrinsic and widely recognized challenges in datasets involving occluded humans. Instead of claiming to eliminate this subjectivity, we explicitly quantified it through a user study on inter-annotator agreement. The resulting agreement scores provide a concrete indication of the level of uncertainty, allowing researchers who use ForestPersons to be aware of the inherent ambiguity in visibility annotations.

In addition, the downstream analyses strongly support the practical utility of visibility labeling. As illustrated in Fig. 6, detection performance consistently declines as visibility decreases. This demonstrates that visibility annotations, despite their heuristic nature, capture systematic variations in task difficulty. In the context of SAR applications where only partial body cues may be available, these attributes provide an indispensable dimension for evaluating the robustness of detection models.

## G  FORESTPERONSIR

Although ForestPersons is composed exclusively of RGB imagery to focus on research challenges central to computer vision, such as complex canopy occlusion patterns, illumination variability, and the visual intricacies of forested environments, infrared (IR) sensing can be highly advantageous in real-world SAR operations. IR sensors offer robustness to such visual complexities, enabling reliable detection of thermal signatures even under severe vegetation occlusion or low-visibility conditions. To reflect this practical relevance, we additionally constructed a dedicated IR dataset tailored for missing-person detection missions.

The IR dataset was captured using a FLIR Boson thermal camera and contains a total of 64,142 images with 79,990 bounding-box annotations. An example of the IR imagery is shown in Fig. 19. The dataset is publicly available at https://huggingface.co/datasets/etri/ForestPersonsIR. It provides temperature-based cues that are difficult to obtain from visible-light imagery, thereby practically complementing the RGB-based ForestPersons for missing person search in forest environments. Finally, we evaluated the baseline object detection models on ForestPersonsIR, as summarized in Table 10.

## H  ZERO-SHOT EVALUATION WITH VISION-LANGUAGE MODELS

To address the rapid advancements in multimodal AI, we expanded our evaluation to include state-of-the-art Vision-Language Models (VLMs). While our primary benchmark focuses on domain-specific object detectors, evaluating modern VLMs provides insight into whether their generalized knowledge can bridge the performance gap in the specific context of under-canopy person detection without fine-tuning.

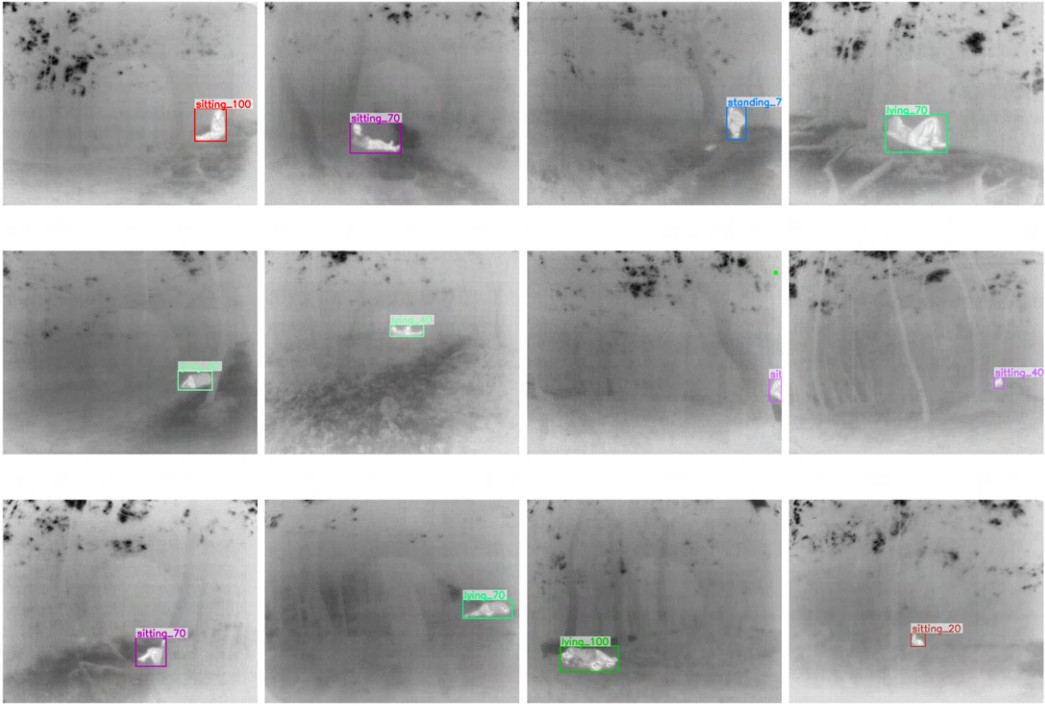

Figure 19: **Visual samples from ForestPersonsIR.** Compared to the RGB images shown in Fig. 3, thermal IR imagery simplifies complex visual patterns and reveals clearer cues for detecting missing persons.

Table 10: **ForestPersonsIR benchmark results.** Object detection model performance on validation and test splits of ForestPersonsIR.

| Detection Model | Validation Split | | | | Test Split | | | |
|---|---|---|---|---|---|---|---|---|
| | AP | AP$_{50}$ | AP$_{75}$ | AR | AP | AP$_{50}$ | AP$_{75}$ | AR |
| YOLOv3 (Redmon & Farhadi, 2018) | 71.2 | 97.4 | 82.7 | 77.7 | 68.9 | 96.4 | 80.8 | 75.3 |
| YOLOXtiny (Ge et al., 2021) | 73.9 | 97.7 | 85.5 | 77.9 | 72.5 | 96.5 | 84.3 | 76.4 |
| YOLOv11n (Jocher & Qiu, 2024) | **84.1** | **99.0** | **96.4** | **86.7** | **74.0** | **98.0** | **86.1** | **79.0** |
| RetinaNet (Lin et al., 2017) | 72.7 | 98.5 | 85.0 | 78.3 | 70.8 | 97.8 | 82.2 | 76.6 |
| Faster R-CNN (Ren et al., 2015) | 72.1 | 97.7 | 84.5 | 76.8 | 71.5 | 97.4 | 83.7 | 75.9 |
| Deformable R-CNN (Dai et al., 2017) | 72.8 | 97.7 | 84.5 | 77.7 | 72.6 | 97.5 | 84.0 | 76.7 |
| SSD (Liu et al., 2016) | 65.1 | 97.2 | 75.5 | 71.6 | 63.2 | 94.5 | 71.2 | 69.8 |
| DETR (Carion et al., 2020) | 64.1 | 96.6 | 75.4 | 73.3 | 61.9 | 95.9 | 69.7 | 71.5 |
| DINO (Caron et al., 2021) | 73.2 | 98.8 | 84.6 | 82.4 | 70.0 | 96.6 | 82.1 | 71.7 |
| CZ Det (Meethal et al., 2023) | 67.8 | 97.3 | 77.7 | 75.3 | 66.8 | 96.9 | 77.7 | 73.7 |

**Experimental Setup.** We categorized the evaluated models into three distinct groups to ensure a comprehensive landscape analysis:

1. **Proprietary MLLMs (Commercial SOTA):** High-performing closed-source models accessed via APIs, known for massive scale and reasoning capabilities. We evaluated Google's Gemini 2.5 series (Comanici et al., 2025) and OpenAI's GPT-4o (Hurst et al., 2024) and GPT-5 OpenAI (2025).

2. **Open-weight MLLMs:** Leading open-source models that allow transparent inference. We selected the Molmo series (Deitke et al., 2025), known for efficient visual grounding, and the Qwen3-VL series (Bai et al., 2025).

3. **Open-vocabulary Object Detectors:** Unlike generative MLLMs, these models are architecturally designed for localization tasks using vision-language alignment. We tested a wide

range of models including OWL-ViT (Minderer et al., 2022), OWLv2 (Minderer et al., 2023), Florence-2 (Xiao et al., 2024), Grounding DINO (Liu et al., 2024), MM-Grounding-DINO (Zhao et al., 2024), and LLMDet (Fu et al., 2025).

**Prompt Engineering for Generative Models.** Since generative MLLMs (Groups 1 and 2) lack explicit object detection heads, we designed a structured prompt to enforce a strictly formatted bounding box output. As illustrated in Figure 20, the prompt instructs the model to act as a detection assistant and return normalized coordinates in a strict JSON format. Responses failing to parse into this schema were discarded as invalid predictions.

---

**Object detection prompt for multimodal large language models**

```
You are an object detection assistant for missing person detection
in forest scenes.
Detect all visible persons in the image.
Return STRICT JSON ONLY. No explanations, no markdown, no comments.
Output format:
{
   "detections": [
      {
         {"bbox": [x_min, y_min, width, height]
      }
   ]
}
Rules:
- bbox is in COCO style: [x_min, y_min, width, height].
- All coordinates are normalized floats in [0, 1]
   relative to the full image width/height.
- Ensure:
   0 <= x_min <= 1,
   0 <= y_min <= 1,
   0 < width <= 1,
   0 < height <= 1,
   x_min + width <= 1,
   y_min + height <= 1.
- If there is no person, return "detections": [].
- Do NOT include any other keys.
- Do NOT wrap the JSON in code fences.
- Do NOT add trailing commas or extra whitespace.
Your output must be valid JSON parsable by Python json.loads.
```

Figure 20: Prompt template used to enforce structured bounding box outputs from generative MLLMs.

**Evaluation Results.** The zero-shot evaluation results on the ForestPersons test set are summarized in Table 11. The experiments reveal a distinct performance divide between generative MLLMs and specialized open-vocabulary detectors.

Generative MLLMs, despite their strong reasoning capabilities, generally struggled with precise localization tasks. Most proprietary models, including GPT-4o, GPT-5 and Gemini 2.5 Flash, exhibited negligible performance, This underperformance stems largely from the architectural discrepancy between their autoregressive text-generation objectives and the precise coordinate regression required for detection, compounded by a lack of domain-specific supervision. An exception was Gemini 2.5 Pro, which demonstrated limited localization capability with an $AP_{50}$ of 11.5%, yet it still fell significantly short of specialized detectors. Nevertheless, this emerging capability suggests that future general-purpose VLMs, with improved spatial alignment, hold the potential to bridge this gap.

A similar trend was observed in open-weight models. The Molmo series failed to produce valid detections, likely due to its training objective on the Pixmo dataset (Deitke et al., 2025), which emphasizes pointing and counting rather than explicit bounding box regression. Furthermore, within the Qwen series, the massive 235B model performed worse than the smaller 8B model. This suggests

Table 11: **VLM detection results on ForestPersons.** Evaluation of vision–language models (VLMs), including closed-weight, open-weight, and open-vocabulary variants, on the ForestPersons test split.

| Detection Model | Test Split | | | |
|---|---|---|---|---|
| | AP | $AP_{50}$ | $AP_{75}$ | AR |
| Proprietary MLLMs | | | | |
| GPT-4o (Hurst et al., 2024) | 0.0 | 0.2 | 0.0 | 0.6 |
| GPT-5 (OpenAI, 2025) | 0.1 | 0.7 | 0.0 | 1.4 |
| Gemini 2.5 Flash (Comanici et al., 2025) | 0.0 | 0.2 | 0.0 | 0.6 |
| Gemini 2.5 Pro (Comanici et al., 2025) | 2.2 | 11.5 | 0.1 | 8.1 |
| Open-weight MLLMs | | | | |
| Molmo 7B-O (Deitke et al., 2025) | 0.0 | 0.0 | 0.0 | 0.0 |
| Molmo 7B-D (Deitke et al., 2025) | 0.0 | 0.0 | 0.0 | 0.0 |
| Molmo 72B (Deitke et al., 2025) | 0.0 | 0.0 | 0.0 | 0.1 |
| Qwen3-VL 8B (Bai et al., 2025) | 5.0 | 21.4 | 0.6 | 14.2 |
| Qwen3-VL 235B (Bai et al., 2025) | 0.0 | 0.1 | 0.0 | 0.7 |
| Open-vocabulary Object Detectors | | | | |
| OWL-ViT (Minderer et al., 2022) | 49.2 | 77.2 | 56.8 | 54.8 |
| OWLv2 (Minderer et al., 2023) | 42.3 | 67.9 | 47.9 | 49.5 |
| Florence2 (Xiao et al., 2024) | 27.3 | 44.0 | 30.2 | 43.9 |
| Grounding-DINO (Liu et al., 2024) | 52.4 | 77.8 | 58.8 | 58.9 |
| MM-Grounding-DINO (Zhao et al., 2024) | 46.1 | 66.4 | 53.5 | 52.1 |
| LLMDet (Fu et al., 2025) | 26.3 | 42.5 | 28.4 | 68.7 |

that scaling up parameters improves semantic generation but does not necessarily translate to better spatial precision or adherence to strict coordinate formatting constraints.

In contrast, open-vocabulary detectors designed explicitly for localization demonstrated significantly better performance. Grounding DINO and OWL-ViT achieved respectable zero-shot scores with an $AP_{50}$ of 77.8% and 77.2%, respectively. However, even the best-performing zero-shot models still lag behind the domain-specific baseline established in our work (Faster R-CNN trained on ForestPersons achieves an $AP_{50}$ of 92.7%). This gap confirms that while modern VLMs offer impressive generalization, domain-specific training remains essential for reliable person detection in complex, occluded forest environments.

# I  TRAINING GENERATIVE MODELS TO CREATE EXTREME SAR SITUATION

We constructed the ForestPersons by capturing volunteers in a controlled environment to simulate missing person scenarios, thereby securing a diverse range of poses and visibility levels. However, due to ethical constraints, collecting data on extreme conditions often encountered in real-world SAR missions (e.g., subjects who are injured, buried, or suffering from hypothermia) remains a challenge.

To address this limitation, our future work involves training generative models based on the Forest-Persons to synthesize missing persons in extreme situations for data augmentation. The objective is to generate high-fidelity synthetic data depicting these atypical distress scenarios to enhance training. We propose this as a practical and scalable approach to bridge the gap between staged data and the stochastic nature of real-world incidents.

Figure 24 presents preliminary examples of synthetic images generated using GLIGEN (Li et al., 2023) finetuned via Dreambooth (Ruiz et al., 2023), demonstrating the feasibility of this proposed direction. The generative model without finetuning produces images that do not fit SAR tasks or in an under-canopy situation, while the generative model finetuned with ForestPersons can produce more realistic images which is fit to SAR tasks in an under-canopy situation.

## J    ELUCIDATING THE EFFECT OF VIDEO CLIP LENGTH

ForestPersons clips were not designed as fixed-length video units; rather, they reflect natural observational opportunities encountered by MAVs operating in dense under-canopy environments. During data collection, factors such as terrain irregularities, heavy vegetation, limited lines of sight, obstacle avoidance, and safety constraints imposed practical limitations on how long continuous sequences could be recorded. Consequently, clip-length variability emerges as a natural characteristic of realistic under-canopy exploration rather than a byproduct of uncontrolled dataset collection.

### J.1    EFFECT OF THE NUMBER OF FRAMES ON DETECTION ACCURACY

To investigate the impact of available temporal information, we conducted an additional analysis by systematically reducing the number of frames sampled from each clip. Faster R-CNN were trained on subsets created by uniformly decreasing the proportion of frames in each sequence, ranging from 100% down to 10%. The results, summarized in Table 12, show that detection accuracy decreases consistently and monotonically as fewer frames are used. This trend indicates that under-canopy environments exhibit substantial variability in visibility, pose, occlusion, and background complexity, and that a richer set of frames provides essential visual diversity for training robust detectors.

Table 12: Performance of Faster R-CNN trained on subsets of frames sampled at different ratios from each clip.

| Ratio of Train Set | 100% | 90% | 80% | 70% | 60% | 50% | 40% | 30% | 20% | 10% |
|---|---|---|---|---|---|---|---|---|---|---|
| AP (Test Split) | 65.3 | 65.3 | 64.4 | 64.9 | 64.2 | 64.2 | 63.6 | 62.5 | 61.6 | 60.8 |

### J.2    IMPACT OF VIEWPOINT DIVERSITY ON DETECTION PERFORMANCE

Dense under-canopy environments inherently restrict the ability to isolate viewpoint as an independent experimental variable, due to constraints related to terrain, vegetation density, visibility, and MAV safety. As a partial proxy for evaluating viewpoint diversity, we conducted an experiment in which the number of training clips was progressively reduced from 100% to 10%. This reduction naturally decreases viewpoint variability as well as scene diversity, such as background structure and occlusion patterns.

As shown in Table 13, detection accuracy drops consistently as clip diversity decreases. Although this experiment does not constitute a controlled multi-view analysis, it provides indirect but meaningful evidence that viewpoint and scene diversity positively influence detection performance in under-canopy conditions.

Table 13: Performance of Faster R-CNN trained on decreasing numbers of training clips.

| Ratio of Train Set | 100% | 90% | 80% | 70% | 60% | 50% | 40% | 30% | 20% | 10% |
|---|---|---|---|---|---|---|---|---|---|---|
| AP (Test Split) | 65.3 | 65.1 | 63.7 | 63.2 | 62.6 | 62.0 | 60.2 | 59.7 | 56.2 | 53.5 |

### J.3    PERFORMANCE SATURATION WITH RESPECT TO THE NUMBER OF TRAINING FRAMES

We further examined whether performance saturates as more frames are used during training. In our dataset, detection accuracy began to plateau when approximately 90% of the available frames were used (roughly 61,000 out of 67,000 images). This behavior reflects the specific environmental and visual characteristics present in the ForestPersons test set. It should not be interpreted as a universal saturation point for all under-canopy scenarios, particularly those involving environmental conditions not represented in the dataset. Thus, the observed 90% saturation is a dataset-specific empirical observation rather than a general claim about optimal training data volume.

Table 14: FPS for each object detection model measured with AMD EPYC 7413 at 640×480 resolution.

| Model | FPS |
|---|---|
| YOLOv3 Redmon & Farhadi (2018) | 6.68 |
| YOLOX Ge et al. (2021) | 13.55 |
| YOLOv11 Jocher & Qiu (2024) | 11.32 |
| RetinaNet Lin et al. (2017) | 1.59 |
| Faster R-CNN Ren et al. (2015) | 0.73 |
| Deformable Faster R-CNN Dai et al. (2017) | 0.31 |
| SSD Liu et al. (2016) | 17.90 |
| DETR Carion et al. (2020) | 2.29 |
| DINO Pan et al. (2025) | 1.06 |
| CZ Det Meethal et al. (2023) | 0.77 |

Table 15: FPS for each object detection model measured with RTX 3090 at 640×480 resolution.

| Model | FPS |
|---|---|
| YOLOv3 Redmon & Farhadi (2018) | 118.00 |
| YOLOX Ge et al. (2021) | 115.86 |
| YOLOv11 Jocher & Qiu (2024) | 104.81 |
| RetinaNet Lin et al. (2017) | 38.12 |
| Faster-RCNN Ren et al. (2015) | 35.02 |
| Deformable Faster-RCNN Dai et al. (2017) | 31.81 |
| SSD Liu et al. (2016) | 76.29 |
| DETR Carion et al. (2020) | 44.17 |
| DINO Pan et al. (2025) | 13.95 |
| CZ Det Meethal et al. (2023) | 16.30 |

## K    REAL-TIME PERFORMANCE ON VARIOUS HARDWARE SPECIFICATIONS

We provide real-time inference performance measurements of representative object detection models across different hardware platforms. To evaluate edge-device feasibility, experiments were conducted on Jetson Orin Nano and Jetson Orin AGX, both configured in MAXN power mode. Among the detectors capable of achieving real-time throughput on these devices, the YOLOv11n model—identified as the best-performing option for onboard deployment—was evaluated at an input resolution of 640×480 over a continuous 5-minute run.

On the Jetson Orin Nano, the PyTorch implementation achieved 32.92 FPS, which increased to 38.44 FPS after TensorRT conversion. On the Jetson Orin AGX, the corresponding values were 35.75 FPS and 31.27 FPS, respectively.

To provide a reference for server-side computation, we conducted additional inference performance tests on both an AMD EPYC 7413 CPU and an RTX 3090 GPU, representing a practical edge-server configuration. Tables 14 and 15 summarize the measured throughput for each evaluated object detection model. As illustrated in Fig. 21, a distinct trade-off between speed (FPS) and accuracy (AP) is evident across most models. Notably, YOLOv11 stands out as an exception, achieving both high accuracy and reasonable inference speeds.

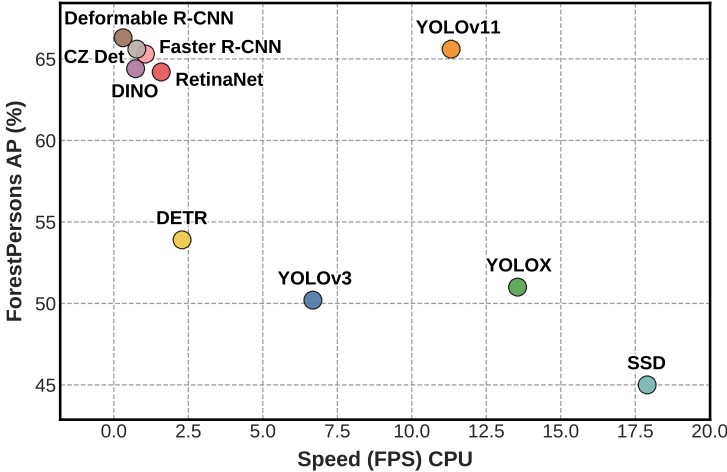

Figure 21: **Speed-accuracy trade-off comparison of various object detection models on the ForestPersons.** The x-axis represents the inference speed in Frames Per Second (FPS), and the y-axis represents the Average Precision (AP).

Table 16: The performance of Faster RCNN models for pose classification tasks.

| Dataset split | mAP | mAP$_{50}$ | mAP$_{75}$ | mAR |
|---|---|---|---|---|
| Validation | 62.0 | 90.8 | 74.7 | 69.5 |
| Test | 56.7 | 84.6 | 66.5 | 67.3 |

## L  EXPERIMENTS ON FORESTPERSONS ATTRIBUTES

### L.1  MULTI-CLASS MISSING PERSON DETECTION ON FORESTPERSONS

ForestPersons includes annotations of the missing person's pose, serving as a proxy for their current physical condition. We anticipate that enabling the automatic assessment of urgency levels, in addition to detecting the presence of missing persons, will significantly advance autonomous SAR operations.

To this end, we extended the object detection task to include pose classification using our collected dataset. Specifically, we trained a Faster R-CNN model to predict the specific posture of the subject (i.e., standing, sitting, or lying). The experimental results are presented in Table 16.

Our experimental analysis shows that training the model for simultaneous person detection and multi-class pose classification leads to decline in average detection precision compared to training solely for binary person detection (person vs. background). This performance trade-off suggests that the additional complexity and classification difficulty introduced by the pose attribute require the model to allocate significant capacity, which can compromise the fundamental bounding box localization performance. This highlights an important area for future multi-task architecture design within the SAR domain.

### L.2  FEATURE FUSION USING CONTEXTUAL INFORMATION

ForestPersons includes not only annotations for the presence of missing persons but also environmental metadata such as season and location. We hypothesize that a detection model aware of these contextual priors could achieve improved performance in SAR scenarios.

Therefore, We have investigated the potential of incorporating conditional information (Weather, Place) as additional context via the simple FiLM structure (Perez et al., 2018) to fuse the context features and the visual features.

Specifically, we integrated this fusion mechanism immediately preceding the detection head. In our design, discrete context labels are first projected into a latent space via an embedding layer. These context embeddings are then used to modulate the visual features through the FiLM layer before they are fed into the detection head for final prediction.

Table 17: Performance of Faster R-CNN models with different contextual inputs using FiLM.

| Model (context) | Validation Split | | | | Test Split | | | |
|---|---|---|---|---|---|---|---|---|
| | AP | AP$_{50}$ | AP$_{75}$ | AR | AP | AP$_{50}$ | AP$_{75}$ | AR |
| Original (no additional context) | 64.2 | 95.6 | 76.5 | 69.6 | 64.4 | 92.7 | 75.4 | 70.0 |
| Weather (FiLM) | 64.4 | 94.6 | 77.0 | 69.4 | 65.2 | 92.9 | 77.4 | 70.2 |
| Place (FiLM) | 64.1 | 94.6 | 76.2 | 69.3 | 65.2 | 93.0 | 76.8 | 70.3 |

The results are shown in Table 17. Note that the validation AP slightly improves upon the test AP compared to the Faster R-CNN performance reported in Table 3. These findings indicate that incorporating additional metadata from the dataset can enhance the performance of object detection models, even when using simple feature fusion methods. Moreover, the results demonstrate that the metadata provided by ForestPersons can contribute to performance improvement when effectively integrated into the model.

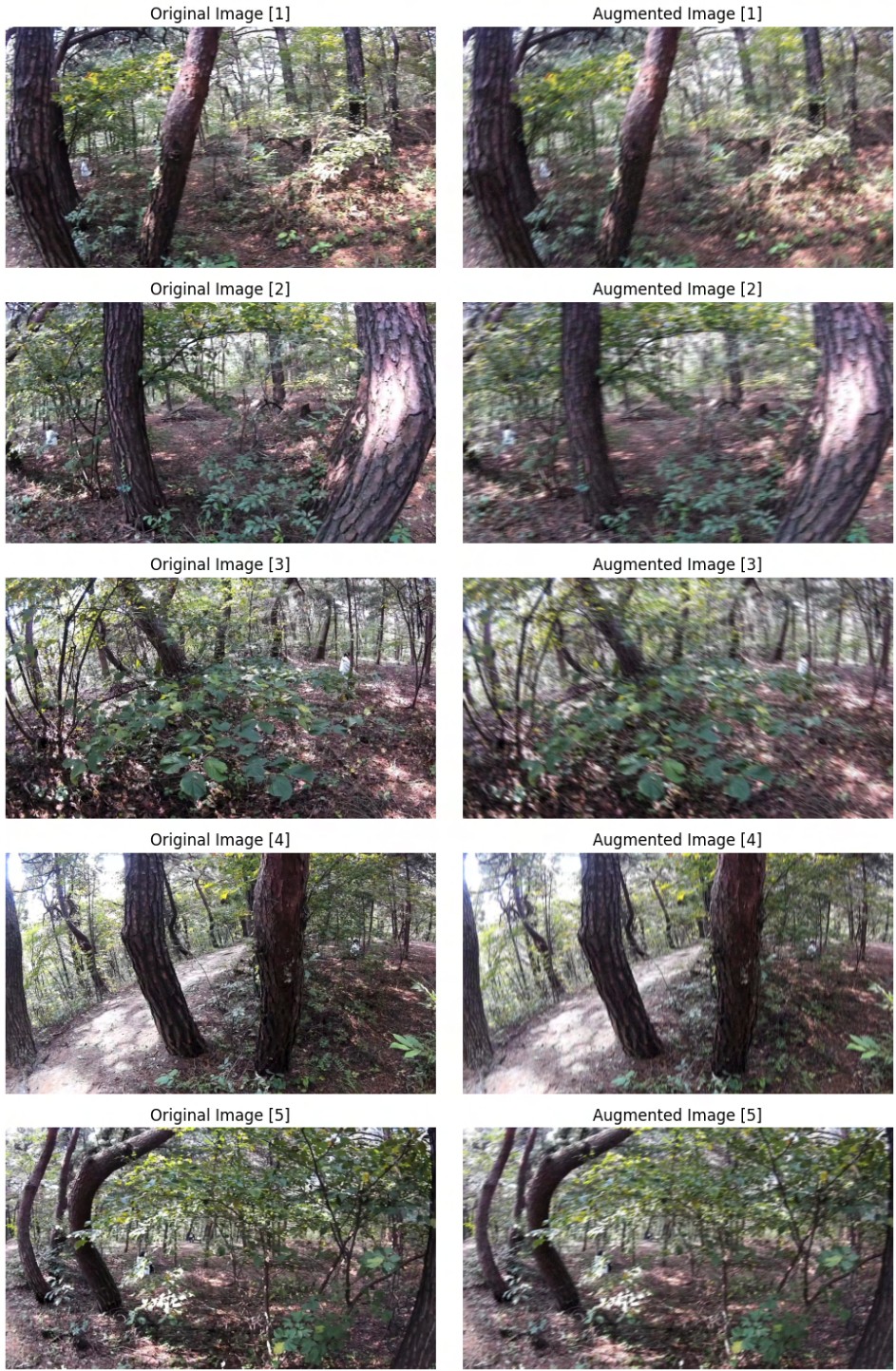

Figure 22: **Examples of ForestPersons augmented by motion blur and sensor noise**. The motion-blur augmentation has the potential to approximate artifacts caused by MAV maneuvers, exposing the model to more realistic motion-induced artifacts.

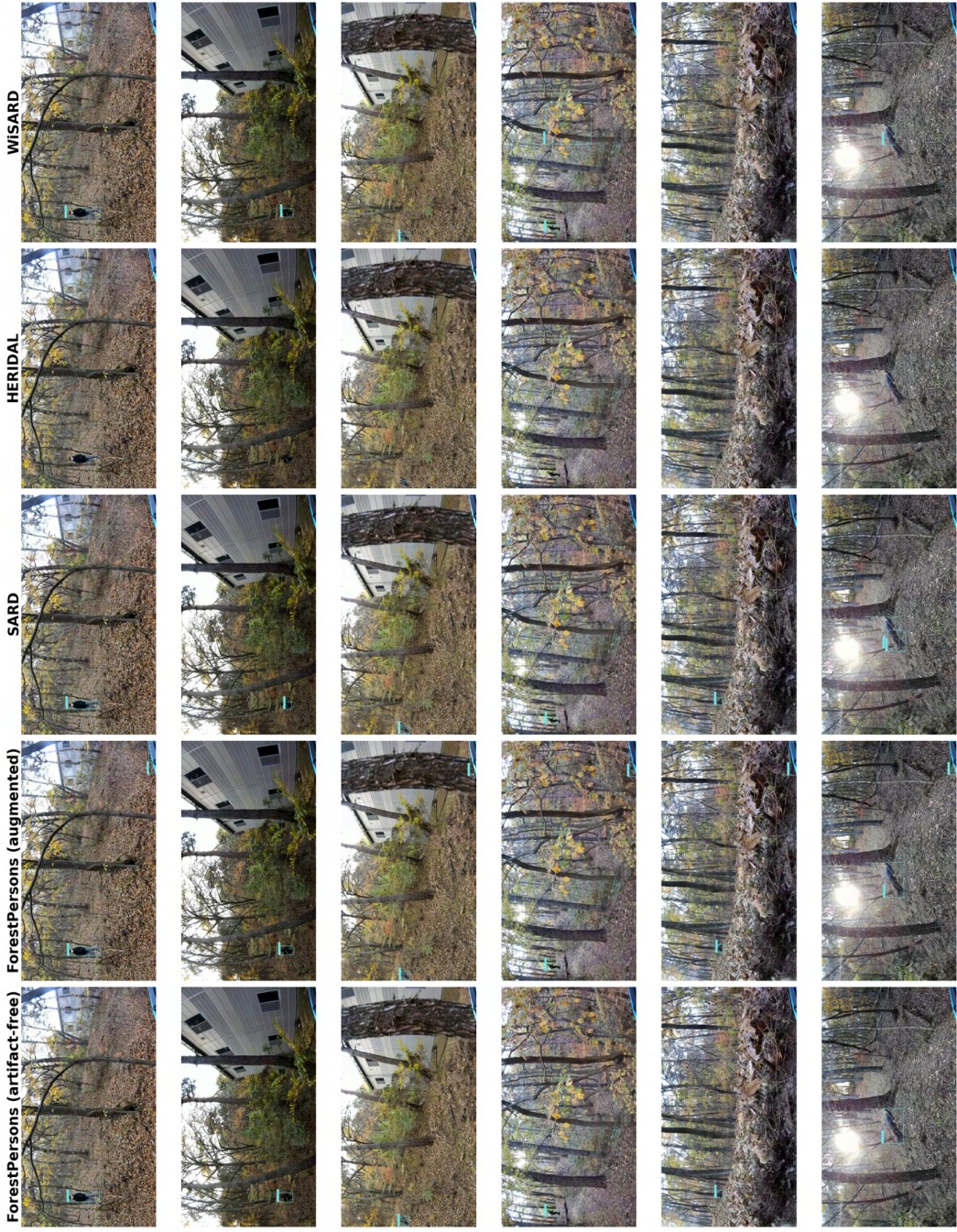

Figure 23: **Predictions of the trained models on the test dataset collected with real MAV.** Faster R-CNN trained on existing SAR datasets exhibit inferior detection performance compared to model trained on ForestPersons, primarily due to the domain gap arising from differences in altitude.

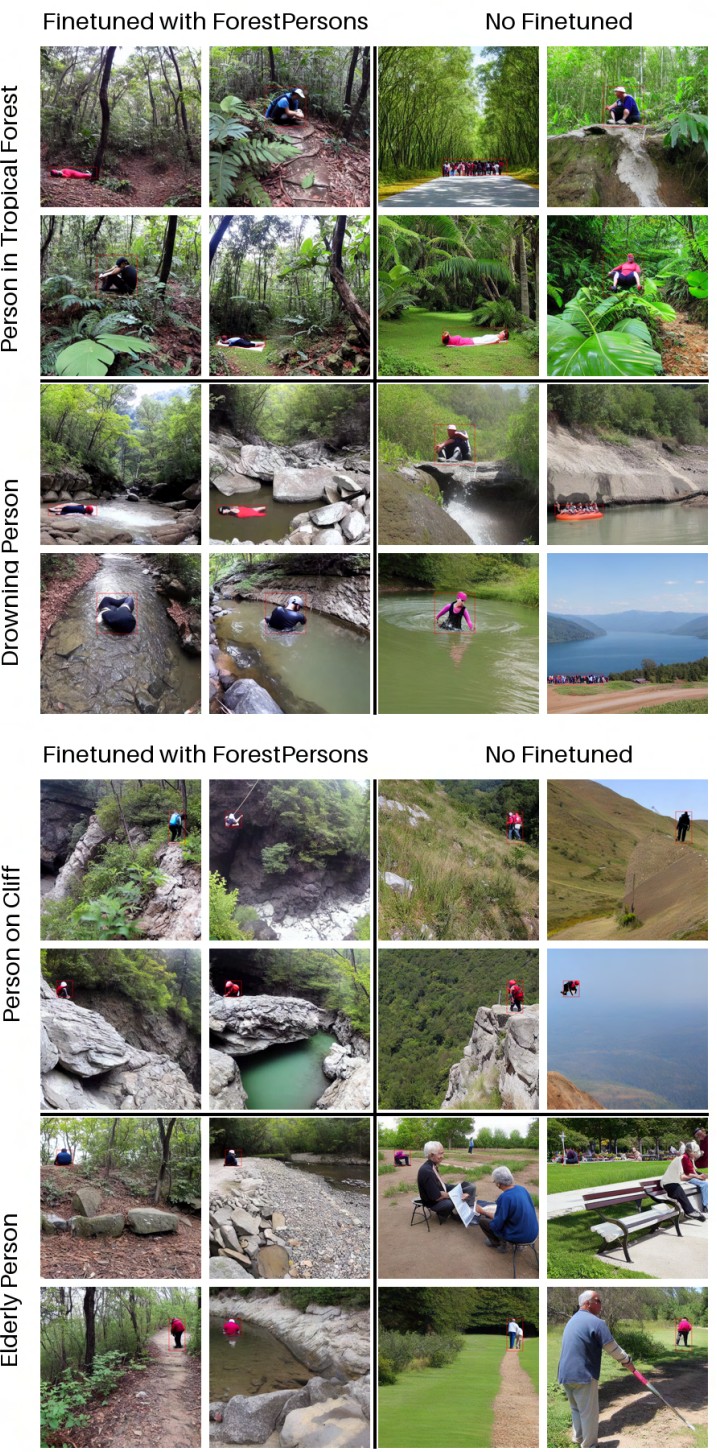

Figure 24: **Synthetic data to depict the extreme situation in the SAR task using generative models.** The red box indicates the bounding box conditioned on the generative model. (Left) The generated images created by generative models finetuned by ForestPersons. (Right) The generated images created by non-finetuned generative models.

## M    USE OF LLMS

We employed large language models (LLMs) solely for polishing the writing. In addition, during the experimental evaluation, we utilized Vision-Language Models specifically to assess zero-shot performance.

