# OpenReview forum: "ForestPersons: A Large-Scale Dataset for Under-Canopy Missing Person Detection"
_ICLR.cc/2026/Conference — ICLR 2026 Poster_

### Official Review · Reviewer_KaHw · 2025-10-27

**Soundness:** 2
**Presentation:** 2
**Contribution:** 3
**Rating:** 4
**Confidence:** 4

**Summary:**

This paper introduces 'ForestPersons', a new large-scale image dataset intended to support Search and Rescue (SAR) missions for missing persons in forested areas. The authors identify the limitations of existing high-altitude drone datasets, where the forest canopy obstructs the view of individuals below. As a solution, they have constructed a dataset of 96,482 images captured by simulating the perspective of Micro Aerial Vehicles (MAVs) at a low altitude of 1.5m–2.0m. This dataset includes diverse seasons (summer, fall, winter), lighting conditions, and poses of the missing person (standing, sitting, lying). A key feature is the quantification and labeling of occlusion levels caused by vegetation as a 'visibility level'.

**Strengths:**

- Clear Problem Definition: The paper successfully highlights the explicit limitations of existing high-altitude SAR datasets (visual obstruction by the canopy) and justifies the need for a specialized dataset focused on the 'under-canopy' environment.

- Data Diversity: The systematic effort to collect data across various seasons (notably including snow-covered winter scenes) and different poses (standing, sitting, lying) to address diverse scenarios in a forest environment is commendable.

- Quantification of Occlusion: A core strength of this dataset is that it does not avoid the key challenge of occlusion in forest environments. Instead, it defines and labels it as a 4-level 'visibility' attribute, providing a benchmark to evaluate model robustness against occlusion.

- Experimental Validation: The authors experimentally demonstrate that models trained on existing datasets (e.g., SARD, COCO) perform poorly on 'ForestPersons' (Table 2), thereby reinforcing the necessity and originality of the proposed dataset.

**Weaknesses:**

1. Unrealistic Simulation of MAV Flight (Domain Gap):
This dataset was not captured by an actual flying MAV, but rather "simulated" by a person holding a camera at 1.5–2.0m. Consequently, unique visual artifacts inherent to actual MAV flight are missing. Specifically, 'Motion Blur' caused by the MAV's rapid movement and vibration, and the 'Rotor Wash' phenomenon, where propeller downwash disturbs surrounding leaves and branches, are not reflected in the data. Therefore, models trained on this dataset may suffer a significant performance drop when deployed on an actual MAV due to this 'Domain Gap'.

2. Limitation of Data Modality (RGB-Only):
This paper relies exclusively on RGB (color) sensor data. However, many state-of-the-art SAR studies, including prior work mentioned by the authors (e.g., WiSARD), adopt a multimodal approach fusing RGB and Thermal imagery as a standard. In real-world forest environments, a missing person may be heavily occluded by bushes or camouflaged, making them impossible to identify with RGB alone. A thermal sensor plays a decisive role in such cases by detecting body heat. The reliance on RGB not only makes night-time detection impossible but also addresses a sub-optimal, simplified version of the real-world problem, as even daytime detection is severely limited.

3. Lack of Realism in Staged Scenarios:
This dataset was built by filming 'actors' performing 'staged missing person scenarios'. Such simulated situations may not adequately reflect the severity and atypical nature of actual distress situations. A real victim may be in a highly irregular pose due to injury or hypothermia, or may be partially buried under dirt, leaves, or debris. The poses in the sample images appear relatively distinct, suggesting a lack of realism in representing data from extreme, real-world scenarios.

4. Subjectivity of Core Annotations:
The reliability of the 'visibility level' label, one of the paper's key contributions, is questionable. The authors' own inter-annotator agreement analysis in Appendix F (Table 8) shows a Cohen's Kappa of approximately 0.45 for the 'visibility' attribute, which statistically represents only "moderate agreement". This implies that the core Ground-Truth label for visibility is highly subjective and noisy. Performance analyses based on this unreliable label (e.g., Figure 6) should be interpreted with caution.

**Questions:**

1. Regarding Unrealistic MAV Flight Simulation (Domain Gap)

Q1.1 (Domain Gap): This dataset was constructed using a handheld camera simulation rather than actual MAV flight. Consequently, dynamic environmental changes unique to MAV flight, such as 'motion blur' or 'rotor wash', are not included in the data. How do you assess the potential impact of this 'Domain Gap' on the performance of a model deployed on an actual MAV?

Q1.2 (Future Plans): To address this 'Domain Gap', do you have plans to collect data using an actual autonomous MAV in the future, or at least to augment the simulation data with realistic noise such as motion blur?

2. Regarding the Limitation of Data Modality (RGB-Only)

Q2.1 (Exclusion of Thermal): In forest SAR environments, thermal sensors play a decisive role in detecting victims occluded by vegetation, as demonstrated in prior studies (e.g., WiSARD) cited in your paper. Could you explain if thermal data was intentionally excluded from your dataset design, or if it was omitted due to collection difficulties?

Q2.2 (Scope Limitation): An RGB-only sensor makes night-time detection impossible and severely limits even daytime detection performance. Despite this limitation, what specific scenarios do you believe this RGB-only dataset can contribute to in real-world SAR operations?

3. Regarding the Lack of Realism in Staged Scenarios

Q3.1 (Realism of Poses): The dataset is built upon 'staged scenarios' featuring 'actors'. However, an actual victim might be in a much more irregular pose due to injury or hypothermia, or partially buried under dirt/leaves, than what is depicted in the sample images (e.g., Figure 3). What are your thoughts on the risk this discrepancy between 'staged' poses and 'actual' victim conditions poses to the model's generalization performance?

4. Regarding the Subjectivity of Core Annotations

Q4.1 (Annotation Reliability): The inter-annotator agreement for the 'visibility level', a key contribution of this paper, was only "moderate" (Cohen's Kappa $\approx$ 0.45, Appendix F). This suggests the label is highly subjective and noisy. Given this low reliability, do you believe the performance analysis based on this label (e.g., Figure 6) is valid and meaningful? We would like to hear your opinion on this.

---

> ### Author Response · Authors · 2025-11-20
> **Author Response to Reviewer KaHw**
>
> We sincerely thank the reviewer for the constructive feedback and for recognizing the _clear problem definition and data diversity of ForestPersons_. We are particularly grateful for the insightful comments regarding domain gaps and data modalities, which have prompted us to conduct additional experiments and release supplementary data.
>
> ---
> **W1, Q1.1, Q1.2. Domain gap (Handheld vs. MAV)**
> 1. We agree that checking the existence of a domain gap between our handheld captured data and the data from actual flying MAV is critical. To address it empirically, we have conducted additional experiments by collecting a new test dataset of 24,209 images using an actual autonomous MAV shown in Fig. 7(a), operating in SAR-relevant forest environments. This new dataset inherently contains the real-world visual artifacts (e.g., occasional motion blur, sensor noise) mentioned by the reviewer. We additionally recorded two individuals in standing, sitting, and lying postures across multiple background settings.
>
> 2. Also, we created an _augmented_ version of the ForestPersons training set by applying transformations that simulate MAV-specific artifacts (e.g., motion blur, sensor noise) using albumentations[5]. We will include visual examples of these augmentations in our revised manuscript.
>
> 3. We evaluated fivemodels: 1) trained on original ForestPersons (referring as _artifact-free_ data), 2) trained on the augmented ForestPersons (referring as _artifact-augmented_ data), 3) trained on SARD, 4) trained on HERIDAL, and 5) trained on WiSARD, on real-world drone test set
>
> 4. The result is shown in Table A. The model trained on our original, artifact-free ForestPersons data achieved a high AP of 0.614 on the new real-world MAV test set. This performance is robust and demonstrates that our handheld data collection method generalizes well to the actual drone domain. Our qualitative analysis of the MAV dataset confirms this, showing that the vast majority of frames are artifact-free, with significant motion artifacts appearing only infrequently during rapid maneuvers.
>
> 5. Conversely, the model trained on the augmented data suffered a severe performance drop (0.358 AP). This strongly suggests that training on data with artificial motion artifacts, when such artifacts are rare in the true target domain, is detrimental. It likely causes the model to overfit to the artifacts themselves rather than the underlying object features, thus harming generalization.
>
> 6. Moreover, we showed that the model trained with existing SAR datasets (SARD, HERIDAL, WiSARD) have lower performance on the MAV test dataset compared to ForestPersons. This is due to the significant domain gap between the high-altitude viewpoint and the low-altitude viewpoint of the dataset, despite the fact that these dataset are captured using MAVs or UAVs. This result validates the effectiveness of our dataset for robust SAR operations in low-altitude, under canopy situations.
>
> 7. Our extensive new experiments validate that the high-quality, artifact-free images from our handheld collection method are not a limitation. Instead, they serve as a robust and effective proxy for training models for real-world MAV deployment, proving to be a more effective training source than data artificially degraded with motion artifacts.
>
> Table A. Model performance comparison between the Faster-RCNN model trained with motion-blur augmented data, and the model trained with original data with respect to real MAV dataset.
>
> |Train dataset|AP|AP$_{50}$|AP$_{75}$|
> |-|-|-|-|
> |ForestPersons (Original)|**61.4**|**88.4**|**76.6**|
> |ForestPersons (Motion blurred)|35.8|64.7|33.9|
> |SARD|23.2|53.5|18.5|
> |HERIDAL|0.0|0.0|0.0|
> |WiSARD|40.2|75.2|35.0|
>
> ---
> **W2, Q2.1, Q2.2.  Limitations of using RGB-only dataset**
> 1. We are pleased to inform that we have, in fact, collected 64,142 images of IR(Infrared) data alongside our RGB dataset. However, we initially excluded this modality from the main benchmark for a specific reason related to annotation quality. We acknowledge the reviewer's insight on the value of multimodal data, and we have decided to release this data to the community. The examples of IR dataset will be contained in the revised manuscript. We commit to publicly releasing this IR dataset on our HuggingFace repository as an extension of ForestPersons as soon as the data curation is finalized.
> ---

---

> ### Author Response · Authors · 2025-11-20
> **Author Response to Reviewer KaHw**
>
> ---
> **W3. Q3.1. Lack of realism in staged scenario**
> 1. We agree that the poses of actual victims in distress may be even more irregular than those of the actors in our dataset.
>
> 2. However, we must respectfully emphasize that collecting data from actual missing persons or placing volunteers in situations of real physical danger (e.g., simulating hypothermia, injury) is ethically prohibitive and practically impossible. As noted in our Ethics Statement, the use of staged scenarios with voluntary participants is the only safe and ethical methodology.
>
> 3. We acknowledge the reviewer's point that extending this to more extreme or atypical poses is a valuable future direction. To address this very challenge, we are exploring the use of generative models finetuned with ForestPersons. Our goal is to create high-fidelity synthetic data depicting these more extreme and atypical distress scenarios, which can be used to augment training. We believe this is a practical and scalable approach to bridge the gap between staged data and the unpredictability of real-world incidents. We will include preliminary examples of these synthetic images in the Appendix of our revised manuscript to demonstrate the feasibility of this future work.
>
> ---
> **W4. Q4.1. Subjectivity of core annotation**
> 1. We emphasize that this challenge is not unique to ForestPersons but is well documented in prior occlusion-focused benchmarks, as we discussed in Appendix F. For example, CrowdHuman[1] requires annotators to complete full-body boxes even when large portions of the person are invisible, COCO-OLAC[2] defines occlusion levels based on visually estimated occlusion ratios, and both HOOT[3] and OVIS[4] depend on human judgment when inferring missing regions or assigning occlusion categories. In all of these cases, the annotation process necessarily involves subjective interpretation because the underlying task—estimating visibility of partially occluded humans—is inherently ambiguous.
>
> 2. Moderate agreement is therefore expected in this setting and reflects the intrinsic difficulty of visibility estimation rather than a flaw in the dataset. Moreover, despite this subjectivity, our visibility labels demonstrate clear empirical utility: as shown in Fig. 6 in the manuscript, all evaluated detectors exhibit a consistent and monotonic decline in performance as visibility decreases. This indicates that the visibility attribute captures meaningful differences in detection difficulty and provides a valid signal for analyzing robustness under natural occlusion.
>
> ---
>     [1] Shao et al., CrowdHuman: A Benchmark for Detecting Human in a Crowd.
>     [2] Wei et al., COCO-OLAC: A Benchmark for Occluded Panoptic Segmentation and Image Understanding.
>     [3] Sahin et al., HOOT: Heavy Occlusions in Object Tracking Benchmark.
>     [4] Qi et al., Occluded Video Instance Segmentation: A Benchmark.
>     [5] Buslaev, et al., Albumentations: Fast and Flexible Image Augmentations.

---

> ### Author Response · Authors · 2025-11-25
> **We have uploaded the revised manuscript**
>
> To address the concerns raised during the initial review, we have incorporated your comments into the revised manuscript.
>
> Specifically, we have added the following sections to address your questions:
> * Domain Gap: Evaluation on a real MAV dataset to further examine the impact of the domain gap between ForestPersons and real SAR environments (Appendix I).
> * Limitations of Using an RGB-only Dataset: Inclusion of examples from an independently collected IR image dataset, which is currently being curated and will be publicly released as an extension of ForestPersons (Appendix G).
> * Lack of Realism in Staged Scenarios: Demonstration of generative model–based synthetic examples as an ethical alternative to filming real missing persons, along with evidence that fine-tuning on ForestPersons improves the realism of generated images (Appendix J).
>
> We appreciate your valuable comments and hope the revisions adequately address your concerns.

---

### Official Review · Reviewer_FMJT · 2025-10-27

**Soundness:** 3
**Presentation:** 2
**Contribution:** 1
**Rating:** 4
**Confidence:** 3

**Summary:**

This paper describes a dataset to benchmark object detection models on the task of detecting humans in a forest from a ground-level perspective. This is intended to improve the performance of below-the-canopy drones in search and rescue (SAR) operations. Most existing benchmarks employ above-the-canopy imagery, which would limit their performance, particularly during leaf-on season.
A suite of object detection models are benchmarked on the proposed dataset.

**Strengths:**

- Paper well written and structured, with correct experimental setup.
- Motivation for the task is made clear.

**Weaknesses:**

1. I appreciate the train-val-test split protocol used in this work, which is done at the sequence level. However, there are not enough details about the potential correlation between sequences. Is it possible that two sequences are captured the same day, on similar locations and with the same subjects? Overall, it would be useful to have some more information about the location of the sequences and the diversity of subjects (to make sure there’s no overfitting to a specific outfit or location type).
2. In the same line, there is no discussion about the potential generalization to other forest types. From the few photos in the paper, it would seem to be some type of temperate broadleaf forest. Although the varying conditions the videos where taken in do suggest the dataset covers a large diversity of environments, I can only imagine that this would hardly work in denser tropical forests. It would be helpful to get an indication of which biomes are covered by the dataset, in order to assess potential for geographical generalization.
3. This paper presents a dataset where the main edge is that it is capture in a different setting that other comparable datasets. As such, there is little novelty to speak of.  Novelty is typically a requirement according to the ICLR reviewer guidelines. I’m not 100% sure what this entails when it comes to datasets, but I would imagine enabling the benchmarking of so far un-benchmarkable tasks. The proposed dataset does not allow to evaluate methods on anything that is fundamentally different, although its different viewpoint and diversity will likely be helpful to train models that will be useful to practitioners. As such, it maybe worth questioning the adequacy of ICLR as a venue to publish this paper, although I do commend the authors for the quality of their work.

**Questions:**

I would like to read the response of the authors to the questions formulated in weaknesses 1 and 2:
- Can they provide statistics about commonalities between video sequences in terms of the location, time and subjects?
- Have the authors considered to which geographical locations they would expect the dataset to generalize to?

**Details Of Ethics Concerns:**

I'm not sure how this could be addressed, but there are clear military applications for this task that I, personally, find a bit concerning.

---

> ### Author Response · Authors · 2025-11-20
> **Author Response to Reviewer FMJT**
>
> We sincerely thank the reviewer for the detailed and constructive feedback. We are encouraged that you found the paper _well-written and the motivation for the under-canopy SAR task clear_. We also appreciate your critical inquiries regarding dataset diversity, generalization scope, and the paper’s fit for ICLR.
>
> ---
> **W1, Q1. Correspondence between the video clips**
> 1. To address the reviewer’s question regarding commonalities between video clips, we provide the following **quantitative statistics that summarize the temporal diversity of our data collection**. ForestPersons was recorded across 22 separate days (locations), ensuring that the dataset reflects a broad range of natural under-canopy conditions rather than repeated captures of the same scene.
>
> 2. **Seasonal diversity**. The dataset spans eight different months, covering both leaf-on and leaf-off periods as well as transitional seasons. The monthly distribution is: January (44 clips), May (39), June (150), July (2), August (28), September (62), October (45), and November (8). This broad seasonal coverage produces variations in foliage density, environmental structure, and illumination.
>
> 3. **Time-of-the-day diversity**. Recordings were conducted between 07:00 and 17:00, resulting in substantial diversity in lighting, shadows, and visibility. The hourly distribution is as follows: 07:00 (22 clips), 08:00 (7), 09:00 (11), 10:00 (16), 11:00 (66), 12:00 (96), 13:00 (52), 14:00 (45), 15:00 (13), 16:00 (33), and 17:00 (17). These variations reduce unintended similarities across sequences and prevent any single lighting condition from dominating the dataset.
>
> 4. **Location diversity**. Each clip was recorded in a distinct forest area or, when captured within a similar region for operational efficiency, the camera and actor were deliberately moved between sequences. This ensured continuous changes in background composition and scene structure, minimizing overlap in visual appearance across video sequences.
>
> 5. Taken together, these statistics demonstrate that ForestPersons incorporates substantial temporal and seasonal variation, thereby minimizing commonalities across video sequences and supporting the robustness of our dataset for benchmarking under-canopy person detection.
>
> ---
> **W2, Q2. Limited biomes (no denser tropical forest)**
> 1. ForestPersons was collected in temperate broadleaf and mixed forests, which represent only a subset of global forest biomes. We agree that models trained solely on ForestPersons may experience additional degradation when applied to much denser environments such as tropical rainforests. However, this limitation does not diminish the value of ForestPersons; rather, it highlights its role as a starting point for broader generalization research across diverse forest environments, including tropical forests.
>
> 2. The core challenges of under-canopy person detection—natural and structural occlusion by vegetation, background clutter, highly irregular textures, and the small visible scale of the person—are not phenomena unique to temperate forests. These structural difficulties are common across many forest biomes, with tropical forests differing primarily in degree rather than in kind. In this sense, ForestPersons captures the fundamental visual obstacles that underlie under-canopy detection problems more generally.
>
> 3. As the first publicly available benchmark that systematically quantifies these challenges in a controlled manner, ForestPersons provides a practical and foundational basis for developing methods that can eventually generalize to more demanding environments, including dense tropical forests. We believe that addressing the occlusion and background-complexity failures exposed by ForestPersons is a necessary precursor to tackling the additional difficulties present in tropical biomes.
> ---

---

> ### Author Response · Authors · 2025-11-20
> **Author Response to Reviewer FMJT**
>
> ---
> **W3. Unsuitability to ICLR dataset section**
> 1. We respectfully but strongly argue that ForestPersons precisely fulfills the definition of novelty cited by the reviewer: enabling the benchmarking of so far un-benchmarkable tasks. As demonstrated in Table 2 in the manuscript, models trained on existing UAV-based SAR datasets (e.g., SARD, HERIDAL) or ground-level datasets (e.g., COCO, CityPersons) fail when applied to the under-canopy domain (e.g., HERIDAL-trained model achieves only 0.2 AP on our test set ).\
> This empirical evidence proves that the task of "under-canopy missing person detection" was effectively un-benchmarkable using prior resources due to the severe domain gap caused by dense occlusion and unique viewpoints. ForestPersons is the first dataset to bridge this gap, establishing a reliable benchmark for this critical real-world problem.
>
> 2. Moreover, unlike other SAR datasets or other human datasets, we provide critical benchmarks for non-standing poses (sitting, lying). Crucially, as discussed in Section 3.2, these pose attributes serve as a vital proxy for the victim's physical state. For instance, a "lying" pose often implies a more critical condition (e.g., unconsciousness, injury) compared to standing. Existing high-altitude datasets cannot reliably capture these subtle postural cues due to limited resolution and top-down perspectives.\
> Therefore, ForestPersons uniquely enables the development of priority-aware SAR systems that can assess urgency, which is a clear and necessary advancement for the field.
>
> 3. To sum up, our dataset unlocks the capability to evaluate and improve machine learning models in a high-stakes domain (under-canopy SAR) that was previously inaccessible. We believe this contribution to robustness, generalization, and real-world application is well-aligned with the scope and interests of the ICLR Datasets & Benchmarks primary area.
>
> ---
> **Ethic concern to military application**
> 1. We want to clearly state that our work is motivated by and dedicated solely to the humanitarian application of Search and Rescue (SAR), aiming to improve autonomous systems for finding missing persons in critical forest environments.
> While we understand that many computer vision technologies carry inherent dual-use potential, we believe the significant positive impact of advancing autonomous technology for saving human lives outweighs hypothetical misuse risks, which we actively discourage through our licensing.
>
> 2. To further ensure responsible use and mitigate such risks, we commit to implementing the following measures:
>
>     (1) **Acceptable Use Policy**: We will provide a detailed Acceptable Use Policy alongside the dataset release. This policy will outline the intended civilian and SAR applications while explicitly prohibiting military, surveillance, or other non-humanitarian uses of the dataset and derived models.
>
>     (2) **License Enhancement**: We will update our License Agreements to contain explicit terms preventing the usage for harm to individuals (including military purposes). To this end, we will refer to established precedents, such as the relevant clauses in the LLAMA 2 Community Licensing Agreement (https://huggingface.co/meta-llama/Llama-2-7b-hf, or similar established industry standards), as a model for our use restrictions.
>
> 4. We believe these measures demonstrate our commitment to ethical compliance and responsible dissemination of research.

---

> ### Author Response · Authors · 2025-11-25
> **We have uploaded the revised manuscript**
>
> We hope that the comments we have provided sufficiently address your concerns.
> Regarding the concern about potential military applications, we have additionally included a statement in the Ethics section of the revised manuscript clarifying that harmful or military use is prohibited. We also plan to explicitly include this clarification in the ForestPersons license on Hugging Face.

---

### Official Review · Reviewer_stPN · 2025-11-01

**Soundness:** 4
**Presentation:** 4
**Contribution:** 4
**Rating:** 8
**Confidence:** 4

**Summary:**

This paper presents a dataset on under-tree canopy people detection, which is different from the over-canopy person detection datasets. Therefore, it contributes a significant new novel step forward in its domain, no dataset like this exists already. The data is big with a lot of varying background conditions. They provide a detailed benchmark with various backbones and also give an analysis on the possible weaknesses of this dataset as well  (inter-annotator dis-/ agreements, failure cases, recall etc). Also, the dataset is available on huggingface and I have downloaded and probed around in it to verify that it has the contents and labels that it claims.

**Strengths:**

Well written and thorough benchmarks with different levels of difficulty and settings.

**Weaknesses:**

The length of the clips is highly variable, between 50-450 frames. An analysis of how the number of frames available vs the person detection accuracy is needed. Does collecting more data on a scene from various angles help get better performance? What is the ideal number of frames, after which the gains are minimal?

One thing I don't feel comfortable is the pose classification. On an initial read, it feels  like they are providing actual human pose instead of what they have provided: lying down, sitting, and standing classifications.

**Questions:**

Is it possible to get an above canopy and a below canopy view, i.e. fly two different drones at the same time ? This could open generative applications, i.e. generating under canopy views from over-canopy views? Something like what was done in the AG-Reid.v2 dataset https://arxiv.org/pdf/2401.02634?

There has been a lot of interest in having fast FPS processing, especially on edge devices. It is not necessarily related to the quality of your dataset, but since you have already done experiments with multiple methods and backbones on different hardware, it will be interesting to see a column/section on the FPS of these various methods? Especially a FPS (or model size or compute time/memory requirement) vs accuracy and uncertainty?

Any experiments on if data attribute (pose, weather, location type,  weather etc) prediction or making the model aware of the attributes helps with improving person detection performance?

---

> ### Author Response · Authors · 2025-11-20
> **Author Response to Reviewer stPN**
>
> We sincerely thank the reviewer for the positive assessment and constructive feedback. We are encouraged by the reviewer's recognition of our dataset as a _significant new novel step forward_ with excellent soundness and contribution. We address the specific comments below.
>
> ---
> **W1. Elucidating the effect of length of video clip**
>
> 1. **Highly variable clip lengths**. Clips in ForestPersons were not designed as fixed-length video units; rather, they reflect the natural observational opportunities available to MAVs operating under dense under-canopy conditions. During data collection, factors such as terrain irregularities, dense vegetation, limited lines of sight, obstacle avoidance, and safety constraints significantly restricted how long continuous sequences could be captured. As a result, clip-length variability emerges naturally from these operational constraints and should be viewed as an inherent characteristic of realistic under-canopy exploration, not as a lack of control in dataset construction.
>
> 2. **Effect of the number of frames on detection accuracy**. To address the reviewer’s suggestion, we conducted an additional analysis by systematically reducing the number of frames sampled from each clip. Detectors were trained on subsets formed by uniformly decreasing the proportion of frames in each sequence, starting at 100% and ending at 10%. The results are in Table A, indicating that detection accuracy decreased consistently and monotonically as fewer frames were used. This indicates that under-canopy environments exhibit substantial variability in visibility, pose, and background complexity, and that a richer set of frames provides important visual diversity for learning robust detectors.
>
> Table A. Performance of the Faster R-CNN trained on various subsets of frames sampled at different rates from each clip.
> |Ratio of train set|100%|90%|80%|70%|60%|50%|40%|30%|20%|10%|
> |-|-|-|-|-|-|-|-|-|-|-|
> |AP (test set)|65.3|65.3|64.4|64.9|64.2|64.2|63.6|62.5|61.6|60.8|
>
> 3. **Impact of viewpoint diversity on detection performance**. We agree that observing a scene from multiple viewpoints can improve robustness. However, in dense under-canopy environments, isolating viewpoint as an independent experimental variable is practically infeasible due to constraints imposed by terrain, vegetation, visibility, and MAV safety. To partially address the reviewer’s concern, we conducted a proxy experiment by reducing the number of training clips (100% → 10%), which naturally reduces viewpoint, background, and occlusion diversity. As depicted in Table B, reducing clip diversity led to a clear and measurable drop in accuracy. Although not a pure multi-angle experiment, this provides indirect but meaningful evidence that viewpoint/scene diversity contributes positively to performance in under-canopy detection tasks.
>
> Table B. Performance of the Faster R-CNN trained on various number of clips
> |Ratio of train set|100%|90%|80%|70%|60%|50%|40%|30%|20%|10%|
> |-|-|-|-|-|-|-|-|-|-|-|
> |AP (test set)|65.3|65.1|63.7|63.2|62.6|62.0|60.2|59.7|56.2|53.5|
>
> 4. **Performance saturation with respect to the number of training frames**. We further examined whether detection accuracy saturates as more frames are included. In our dataset, using approximately 90% of the available images (~61,000 out of 67,000 frames) resulted in the performance beginning to plateau. We emphasize, however, that this saturation pattern reflects the specific environmental conditions present in our test set and should not be interpreted as a universal saturation point for all under-canopy scenarios—particularly those not represented in ForestPersons. Thus, the observed 90% plateau is a dataset-specific empirical observation, not a general claim about optimal data volume for under-canopy detection.
>
> ---
> **Q1. Collecting pairs of above-canopy and under-canopy views**
> 1. We appreciate the reviewer’s suggestion regarding the possibility of acquiring paired above- and under-canopy views, as well as the potential generative applications it might enable. This is indeed an interesting direction, particularly for tasks such as estimating under-canopy density or navigational difficulty from safer above-canopy viewpoints, which could support MAV flight planning.
>
> 2. However, in the dense under-canopy SAR environments considered in our work, forest canopy is typically closed enough so that no visual cues of missing persons—or even the ground surface—are observable from above the canopy. As a result, ForestPersons focuses exclusively on under-canopy views, where visual detection is feasible.
>
> 3. While cross-view generative modeling between above- and under-canopy observations may be possible in principle for certain tasks, the absence of any person-related visual information in the above-canopy view suggests that generating missing-person–conditioned under-canopy imagery from overhead observations would be extremely challenging, and likely infeasible in the first place.
> ---

---

> ### Author Response · Authors · 2025-11-20
> **Author Response to Reviewer stPN**
>
> ---
> **Q2. Measuring FPS on various hardware spec**
> 1. We agree with the reviewer that reporting real-time performance for the benchmarked models—independent of the dataset itself—would be useful for researchers interested in deploying detectors on resource-constrained platforms. To this end, we conducted experiments on Jetson Orin Nano and Jetson Orin AGX—edge devices that can be realistically mounted on MAVs—both operated in MAXN power mode.
>
> 2. Using YOLOv11n, the only model among the benchmarked detectors that achieves real-time throughput, we measured inference speed at an input resolution of 640×480 over a continuous 5-minute run. PyTorch implementation achieved 32.92 FPS on Orin Nano and 35.75 FPS on Orin AGX. After TensorRT conversion, the throughput increased to 38.44 FPS on Orin Nano and 31.27 FPS on Orin AGX.
>
> 3. To provide reference for the server-side component, we additionally measured the inference performance on an RTX 3090, which represents a realistic edge-server configuration. The throughput values are shown in Table C. Note that the performance and FPS of the model are inversely related.
>
> Table C. The FPS for each object detection model measured with RTX 3090
> |RTX 3090 (640x480)|YOLOv3|YOLOX|YOLOv11|RetinaNet|Faster-RCNN|Deformable Faster-RCNN|SSD|DETR|DINO|CZ Det|
> |-|-|-|-|-|-|-|-|-|-|-|
> |FPS |118.00|115.86|104.81|38.12|35.02|31.81|76.29|44.17|13.95|16.30|
>
> ---
> **W2. Lack of providing various human poses instead of categorical labels**
> 1. We intended to provide classifications of the critical physical state of the missing person rather than a fine-grained geometrical pose description (like keypoints). The categories (Standing, Sitting, Lying) are designed to serve as crucial semantic cues regarding the individual's physical state or urgency (Triage) . As we noted in Section 3.2, a 'lying' posture often implies a more critical condition, such as unconsciousness or injury, requiring higher priority in SAR missions.
>
> 2. In fact, we found that collecting high-quality keypoint annotations for fine-grained pose was technically infeasible due to the severe and unpredictable occlusion caused by dense vegetation in the under-canopy environment. Keypoint annotation standards rely on visible body joints, which are frequently hidden in our challenging scenes, leading to excessive noise to the labels.
>
> 3. Therefore, we focused on providing the most valuable information for SAR practitioners: a categorical proxy for the victim's physical status, which is both robust to annotate and directly relevant to mission prioritization.
>
> ---
> **Q3. Experiment for pose classification using data attributes**
> 1. The results for the multi-task model trained with pose classification are summarized below:
>
> Table D. The performance of Faster RCNN models for pose classification tasks.
> |Dataset split|AP|AP$_{50}$|AP$_{75}$|
> |-|-|-|-|
> |Validation|0.620|0.908|0.747|
> |Test|0.567|0.846|0.665|
>
> 2. Our experimental analysis shows that training the model for simultaneous person detection and multi-class pose classification leads to decline in average detection precision compared to training solely for binary person detection (person vs. background).
>
> 3. This performance trade-off suggests that the additional complexity and classification difficulty introduced by the pose attribute require the model to allocate significant capacity, which can compromise the fundamental bounding box localization performance. This highlights an important area for future multi-task architecture design within the SAR domain.
>
> ---
> **Q4. Making the model aware of the attributes**
> 1. We thank the reviewer for proposing this constructive experiment. We have investigated the potential of incorporating conditional information (Weather, Place) as additional context via the simple FiLM structure [1] to fuse the context features and the visual features.
>
> 2. The result is depicted in Table E. Note that the validation AP slightly improves test AP compared to the performance of Faster R-CNN depicted in the manuscript.
>
> Table E. The performance of Faster R-CNN models using FiLM structure for feature fusion.
> (Top) No additional context (Mid) Weather as additional context (Bot) Place as additional context.
> | Model (context)|Split|AP|AP$_{50}$|AP$_{75}$|
> |-|-|-|-|-|
> |Original (no additional context)|val|64.2|95.6|76.5|
> |Original (no additional context)|test|64.4|92.7|75.4|
> |Weather (FiLM)|val|64.4|94.6|77.0|
> |Weather (FiLM)|test|65.2|92.9|77.4|
> |Place (FiLM)|val|64.1|94.6|76.2|
> |Place (FiLM)|test|65.2|93.0|76.8|
>
> 3. The results shows that combining the additional metadata of the dataset can improve the performance of object detection model even with the simple feature fusion methods. ForestPersons provides additional various metadata with respect to the dataset, which may have room for improving the performance by combining the metadata to the model.
>
> ---
>     [1] Perez, Ethan, et al., FiLM: Visual reasoning with a general conditioning layer.

---

> > ### Comment · Reviewer_stPN · 2025-11-21
> >
> > Will you include these new results in the final paper? maybe the supplementary?

---

> > > ### Author Response · Authors · 2025-11-21
> > >
> > > Yes, absolutely. We will include these additional experimental results in the Appendix of the revised manuscript. We are finalizing the document and will upload the revision shortly.

---

> ### Comment · Reviewer_stPN · 2025-11-21
>
> Can you also do an inference speed on CPU? I was hoping to see a plot like Fig.1 of this paper below. Please note that I am not asking you to train a model. Just use an already trained model but run it on a CPU.
>
> https://openaccess.thecvf.com/content/WACV2025/papers/Zaveri_Improving_Accuracy_and_Generalization_for_Efficient_Visual_Tracking_WACV_2025_paper.pdf

---

> ### Author Response · Authors · 2025-11-25
> **We have uploaded the revised manuscript**
>
> We sincerely thank you for your consistent and constructive feedback. Following your suggestion, we conducted FPS measurements for various object detection models on CPU and generated an FPS-Performance trade-off figure, by referencing the paper you mentioned.
>
> We have incorporated these results into the revised manuscript. Specifically, we have added the following sections to address your questions:
> * Ablation Study on Video Clip Length: An analysis of the effects of frame count and viewpoint diversity on detection performance (Appendix K).
> * Inference Speed Analysis: Comprehensive FPS measurements on various hardware configurations **(including CPUs)**, along with an **FPS-Performance trade-off comparison** (Appendix L).
> * Attribute-Aware Experiments: Additional experiments investigating the impact of utilizing various metadata attributes (e.g., pose, weather) during training (Appendix M).
>
> We hope these updates satisfactorily address your comments. Thank you again for helping us improve our paper.

---

### Official Review · Reviewer_DXNw · 2025-11-02

**Soundness:** 2
**Presentation:** 3
**Contribution:** 2
**Rating:** 2
**Confidence:** 4

**Summary:**

The paper introduces ForestPersons, a large-scale dataset for under-canopy missing person detection in Search and Rescue (SAR) missions, containing 96,482 images and 204,078 annotations collected from ground-level perspectives to simulate Micro Aerial Vehicle viewpoints. The dataset captures individuals in diverse poses (lying, sitting, standing) who are naturally partially occluded by vegetation, branches, and terrain across different seasons, weather conditions, and lighting. Individuals were positioned in different postures and naturally partially occluded by vegetation, branches, or uneven terrain, with annotations including bounding boxes, pose labels, and visibility levels. Experiments demonstrate large performance drops when applying models trained on existing SAR datasets to ForestPersons, and significant degradation from ground-level person datasets, establishing the need for domain-specific training data.

**Strengths:**

The experimental validation demonstrates a substantial practical problem, with existing SAR models showing catastrophic performance drops on under-canopy scenarios, providing strong empirical evidence for the dataset's necessity and filling a genuine gap in "Search and Rescue" applications that could have real-world impact for missing person detection.

**Weaknesses:**

-- limited technical and scientific novelty: This is mainly a domain-specific dataset contribution without methodological innovations in computer vision or machine learning. The work involves training standard object detection models on forest imagery and demonstrates expected domain transfer limitations, offering no new architectures, techniques, or fundamental insights beyond data collection for a specific application scenario.

-- narrow scope and generalizability: The dataset addresses a very specific task (under-canopy person detection for SAR) with limited broader applicability to computer vision research. While the data collection required significant effort and funding, the contribution is primarily valuable to SAR practitioners rather than advancing general object detection, occlusion handling, or robustness techniques that could benefit the wider research community.

-- simulated rather than authentic data: The dataset uses staged photography with handheld/tripod cameras positioned at 1.5-2m height to simulate MAV perspectives, rather than actual drone footage from real SAR missions, raising questions about ecological validity and whether the simulated conditions truly represent operational SAR scenarios.

**Questions:**

Have you evaluated modern Large Vision-Language Models like GPT-5/4o, Gemini, or Claude on your benchmark? (also open source models like MOLMO?) These models often demonstrate strong zero-shot object detection and description capabilities across diverse domains. Given that your evaluation focuses on traditional object detection models (YOLO, Faster R-CNN, etc.) mostly from 2015-2021, it's unclear whether the identified performance gaps persist with state-of-the-art VLMs that might already handle under-canopy person detection effectively without domain-specific training.

---

> ### Author Response · Authors · 2025-11-20
> **Author Response to Reviewer DXNw**
>
> We sincerely thank the reviewer for the detailed feedback. We respectfully note that while the reviewer expressed concerns regarding novelty and scope, we would like to **clarify the positioning of our work** within the primary area of the ICLR Datasets & Benchmarks. We address these concerns below.
>
> ---
> **W1. Limited technical and scientific novelty**
> 1. The primary aim of this study is to provide a high-quality dataset and a clearly defined problem setting grounded in real search-and-rescue (SAR) operations, in line with the purpose of the ICLR primary area “Datasets and Benchmarks.” Rather than proposing a new model architecture, our goal is to establish a reliable data resource for under-canopy person detection and to show that object detectors for person detection trained on widely used public datasets fail in under-canopy environments.
>
> 2. As acknowledged by the reviewer in the strengths section, our main contribution is the first quantitative demonstration that widely used object detectors for person detection suffer severe performance degradation when deployed under forest canopies (refer to Table 2 in the manuscript). This demonstrates that existing benchmarks inherently lack or insufficiently represent the visual characteristics that define structured natural environments such as forests. By revealing this gap, our study highlights the need for data-centric efforts that capture the true variability of under-canopy environments.
>
> 3. To facilitate reproducibility and further research on this important problem, we publicly release the ForestPersons dataset for the community.
>
> ---
> **W2. Narrow scope of generalization**
> 1. Although ForestPersons was initially motivated by the lack of under-canopy training data for Search and Rescue (SAR) scenarios, we argue that its value extends beyond missing-person detection. It provides a natural-environment benchmark that exposes fundamental limitations in existing datasets and enables systematic investigation of core challenges in computer vision, including occlusion robustness, generalization under distribution shift, and detection in cluttered natural scenes.
>
> 2. First, ForestPersons captures natural and structurally induced extreme occlusion that is rarely represented in existing person-detection datasets. Vegetation, branches, and uneven terrain create complex partial-visibility patterns that differ substantially from those in urban or roadway environments. Our visibility level annotations systematically characterize these varying degrees of observability, enabling analysis and modeling of detection behavior under severe natural occlusion, which is an underexplored yet long-standing challenge in computer vision.
>
> 3. Second, the consistent performance degradation of object detectors in the Table 2 of the manuscript, despite being trained on large-scale public datasets, reveals a potential limitation of current benchmarks. This issue is not specific to the SAR domain; rather, it provides quantitative evidence that widely used datasets may overestimate model generalization by failing to capture the visual complexity, environmental variability, and background structure inherent to natural settings. ForestPersons thus serves as a benchmark for evaluating and diagnosing these generalization gaps.
>
> 4. Third, ForestPersons contains scene-level characteristics that are difficult to examine in existing person-detection datasets but are broadly relevant to the computer vision community. Under-canopy scenes frequently include small-scale human instances, strong occlusion, dense vegetation, irregular textures, and visually ambiguous boundaries. These conditions introduce challenges such as fine-grained localization, human–background confusion, and extreme appearance variability—issues that arise in many real-world deployments beyond SAR. ForestPersons therefore provides a complementary benchmark for studying detection performance in complex, cluttered natural environments.
> ---

---

> ### Author Response · Authors · 2025-11-20
> **Author Response to Reviewer DXNw**
>
> ---
> **W3. Simulated rather than authentic data**
> 1. We understand your question as involving two related aspects: (1) the realism of assuming a 1.5–2 m MAV flight height, and (2) whether staged ground-level photography can meaningfully approximate MAV viewpoints in under-canopy conditions. We address both points below.
>
> 2. First, the 1.5–2 m viewpoint is consistent with realistic SAR operations in _dense under-canopy environments_. In such settings, high-altitude MAV imagery is ineffective because the forest canopy fully obstructs any line of sight to the ground. As documented in SAR practice, the only viable options for locating a missing person are: (i) ground teams physically entering and searching the forest, or (ii) deploying MAVs that navigate below the canopy layer as a substitute or supplement to human searchers. In these scenarios, MAVs must operate at low altitudes precisely because vegetation density, terrain undulation, and obstacle distribution constrain safe flight envelopes to approximately human height. Thus, the viewpoint represented in ForestPersons reflects a realistic and common operational configuration for under-canopy MAV deployment.
>
> 3. Second, collecting large-scale authentic under-canopy MAV imagery containing real missing persons is not feasible due to ethical, legal, and safety constraints. Even with actors, repeatedly flying MAVs in dense vegetation to capture controlled variations of pose, occlusion, viewpoint, and lighting would require excessive cost and time, and would introduce non-negligible safety risks (e.g., MAV collisions near actors). Importantly, staged ground-level capture allowed us to systematically vary visibility levels, human poses, clothing, foliage density, seasonality, and background complexity—factors that are essential for building a benchmark but cannot be adequately controlled or balanced using opportunistic real-mission MAV footage. This approach enabled the creation of a dataset with a well-distributed range of occlusion and appearance conditions, which is necessary to study detection robustness in a principled and reproducible manner.
>
> 4. To further address the reviewer’s concerns about ecological validity, we additionally collected a separate evaluation set comprising 24,209 images from real MAV flights conducted in dense under-canopy environments. We evaluated detectors trained on ForestPersons using this authentic MAV test set. The results persist in real MAV imagery, providing empirical evidence that our dataset meaningfully reflects the challenges of practical SAR deployments. For more detailed experimental settings, please refer to the rebuttal of W1 of Reviewer KaHw.
> ---

---

> ### Author Response · Authors · 2025-11-20
> **Author Response to Reviewer DXNw**
>
> ---
> **Q1. Zero-shot evaluation using recent state-of-the-art VLMs**
>
> We thank the reviewer for raising this critical question. We agree that recent large vision-language models (VLMs) have shown impressive zero-shot capabilities. To address this, we conducted a comprehensive evaluation using three categories of state-of-the-art models to verify whether the performance gap persists without domain-specific training.
>
> **Evaluation Setup.** To provide a systematic analysis representing the current landscape of multimodal AI, we evaluated zero-shot performance on the ForestPersons test set across three distinct model categories:
>
> **Proprietary MLLMs (Commercial SOTA)**. These encompass the highest-performing Multimodal Large Language Models currently available, typically accessed via restricted APIs provided by major tech companies. These models are known for their massive scale and superior reasoning capabilities. (e.g. Google’s Gemini-2.5 series[1] and OpenAI’s GPT series[2,3]).
>
> **Open-weight MLLMs (Accessible Research Models)**. This category includes SOTA open-source models where weights are publicly available, allowing for local deployment and transparent inference. These models have recently narrowed the performance gap with proprietary models. We selected leading models such as DeepSeek's Qwen3-VL series[4] and Allen AI's Molmo[5], running them in a controlled local environment to verify their capabilities.
>
> **Open-vocabulary Object Detectors (Detection Foundation Models)**. Unlike the generative MLLMs mentioned above, these models are fundamentally architected for localization tasks. They act as "foundation models" for object detection, leveraging vision-language alignment to perform zero-shot detection based on text prompts. We employed widely recognized models such as OWL-ViT[6], Florence-2[7], and Grounding DINO[8] to benchmark their effectiveness against generative approaches.
>
> For generative MLLMs, which lack explicit detection heads, we designed a structured prompt to enforce a bounding box output format (details in Appendix H of our revised manuscript). The generated responses were parsed, and predictions were evaluated only if they adhered to the valid coordinate format.
>
> **Results and Analysis**. The results are summarized in Table A below.
>
> Table A. Zero-shot evaluation of state-of-the-art VLMs on ForestPersons test dataset.
> | Model Type | Model Name | AP | AP$_{50}$ | AP$_{75}$ | AR |
> | :--- | :--- | :---: | :---: | :---: | :---: |
> | MLLMs | Gemini 2.5 Flash | 0.0 | 0.2 | 0.0 | 0.6 |
> | | Gemini 2.5 Pro | 2.2 | 11.5 | 0.1 | 8.1 |
> | | Qwen3-VL 8B | 5.0 | 21.4 | 0.6 | 14.2 |
> | Open-Vocab | OWL-ViT | 49.2 | 77.2 | 56.8 | 54.8 |
> | Detectors | Florence-2 | 27.3 | 44.0 | 30.2 | 43.9 |
> | | Grounding DINO | 52.4 | 77.8 | 58.8 | 58.9 |
>
> 1. **Limitations of General MLLMs**. As shown in Table R1, general-purpose MLLMs (Gemini, Qwen) perform poorly on precise localization (AP $\approx$ 0). While they generate plausible bounding-box responses, they lack specialized regression heads for bounding boxes, often leading to invalid coordinates or extremely low IoU scores.
> 2. **Gap in Open-vocabulary Detectors:** Specialized models like Grounding DINO achieved reasonable zero-shot performance ($AP_{50}$ 77.8). However, they still lag significantly behind the traditional detectors trained on our dataset (e.g., the baseline Faster R-CNN trained on *ForestPersons* achieves $AP_{50}$ of 92.7% which is a significantly higher performance than General MLLMs).
>
> While the zero-shot localization capability of the tested MLLMs (e.g., Gemini 2.5) fell short of expectations, we acknowledge the rapid evolution of this field. We are currently extending our evaluation to include OpenAI’s GPT series and other recent VLMs to capture these latest developments. We share the reviewer's excitement about exploring the true upper bound of zero-shot capabilities and look forward to reporting these updated findings.
>
> ---
>     [1] Comanici, et al., Gemini 2.5: Pushing the Frontier With Advanced Reasoning, Multimodality, Long Context, and Next Generation Agentic Capabilities
>     [2] Hurst, et al., GPT-4o System Card
>     [3] OpenAI, GPT-5 System Card
>     [4] Yang, et al., Qwen3 Technical Report
>     [5] Deitke, et al., Molmo and Pixmo: Open Weights and Open Data for State-Of-The-Art Vision-Language Models
>     [6] Minderer, et al., Simple Open-Vocabulary Object Detection
>     [7] Xiao, et al., Florence-2: Advancing a Unified Representation for a Variety of Vision Tasks
>     [8] Liu, et al., Grounding Dino: Marrying Dino With Grounded Pre-Training for Open-Set Object Detection

---

> ### Author Response · Authors · 2025-11-25
> **We have uploaded the revised manuscript**
>
> We sincerely appreciate the reviewer’s insightful comments and constructive suggestions. We have carefully revised the manuscript to address the concerns raised, and the following updates have been incorporated accordingly:
> * Evaluation on Real MAV Data: To address the concern regarding “simulated rather than authentic data,” we conducted additional evaluations on a real MAV dataset to further analyze the domain gap between ForestPersons and real SAR environments (Appendix I).
> * Zero-shot VLM Evaluation with Additional Models: In line with the reviewer’s recommendations, we expanded our analysis by evaluating representative models such as GPT-4o, AllenAI’s Molmo series, and additional open-vocabulary detectors (e.g., MM-Grounding-DINO, LLMDet) (Appendix H).
>
> Regarding GPT-5, due to the significant inference latency attributed to its high-level reasoning processes, the evaluation is currently in progress. We are making our best effort to complete this evaluation and will incorporate the findings into the final manuscript if the timeline permits.
> We believe these extended experiments provide comprehensive evidence regarding the capabilities and limitations of modern VLMs in this domain.

---

### Author Response · Authors · 2025-11-20
**General Response**

We sincerely thank the reviewers for their thoughtful feedback. We are pleased that all reviewers recognized the importance of addressing the under-canopy search and rescue (SAR) problem, and acknowledged the comparative analysis demonstrating limitations of existing benchmarks and detectors on ForestPersons. We were also encouraged by multiple reviewers who found the paper well structured and appreciated the clarity of its problem formulation and the strength of its experimental design.

At the same time, we understand the concerns raised regarding the use of simulated under-canopy data—specifically, **1) handheld/tripod capture, 2) staged actors, and 3) the suitability of the dataset for ICLR**. We address these common concerns below.

---
**1. Handheld/tripod capture (Domain Gap)**:  The reviewers are commonly concerned about the domain gap between the images captured by drone and those captured by handheld/tripod camera. To address these concerns, we conducted additional experiments focusing on motion-related artifacts caused by real MAV flights. Specifically, **we collected separate video clips by flying a MAV** equipped with a camera, capturing missing-person scenarios during flight, for evaluating models trained on ForestPersons. We also created motion-blurred variants of the training sets and trained detectors accordingly, then compared detector performance on models trained with and without motion-blur augmentation using a test dataset captured by real MAV.

The results (presented in the response to Reviewer KaHw) show that the detector trained without motion blur consistently performs better. This is likely because severe motion blur occurs primarily during aggressive MAV maneuvers, whereas in typical under-canopy scanning operations, such strong motion artifacts arise infrequently. Moreover, models trained with Forestperson show better performance compared to those trained with existing SAR datasets, even though they were collected using MAVs and UAVs. For detailed experimental settings, please refer to the rebuttal of W1 of Reviewer KaHw.

---
**2. Staged actors (Realism)**: Several reviewers raised a valid concern regarding the potential lack of realism in our staged scenarios, noting that real victims may present in more irregular poses (e.g., due to injury or hypothermia) than those captured by actors. We agree that the diversity of poses in actual distress situations is crucial for future improvement, but we respectfully emphasize that collecting data from real missing persons or placing volunteers in situations of actual physical danger (e.g., simulating extreme conditions) is **ethically prohibitive**.

To proactively address the demand for more extreme poses in a scalable and ethical manner, we are exploring the use of generative models finetuned with ForestPersons. We believe this represents a practical and scalable approach to bridge the gap between staged environments and real-world unpredictability. We will include preliminary examples of the synthetic images in the Appendix of our revised manuscript to demonstrate the feasibility of this essential future work.

---
**3. ICLR venue suitability**: We emphasize that our work’s objective aligns with the primary area of the ICLR Datasets & Benchmarks. Our goal is not to propose a new model architecture, but to provide a high-quality dataset and to formalize an under-canopy SAR problem grounded in real operational constraints. Prior ICLR-accepted works[1, 2, 3, 4] that provided the data necessary for enabling learning-based researches to address important yet underexplored problems were **consistently well-received at the venue**.

In the same spirit, ForestPersons introduces an unaddressed SAR under-canopy challenge and delivers the first dataset that alleviates the field’s major bottleneck—data scarcity—thereby enabling reproducible and quantitative follow-up research. Beyond domain-specific data collection, ForestPersons reveals that widely used object detection benchmarks overlook critical factors necessary for detection in natural environments, pointing to a gap between current benchmark assumptions and real under-canopy conditions. By publicly releasing the dataset, we provide a foundational resource for a socially consequential yet understudied problem in AI community.

---
With the reviewer’s valuable reviews, in the revised manuscripts, we will have carefully incorporated the additional experimental results, synthetic data examples, and necessary clarifications into the manuscript to address these points comprehensively.

---
    [1] TAU-106K (Dataset for understanding traffic accident) https://openreview.net/forum?id=Fb0q2uI4Ha
    [2] AIMS.au (Dataset for analysis of modern slavery countermeasures) https://openreview.net/forum?id=ybfmpJiKXX
    [3] SWEb (Web dataset for Scandinavian languages) https://openreview.net/forum?id=vhPE3PtTgC
    [4] CircuitNet 2.0 (Dataset for chip design) https://openreview.net/forum?id=nMFSUjxMIl

---

### Author Response · Authors · 2025-11-25
**We have uploaded the revised manuscript**

Dear Reviewers,

We are pleased to announce that we have uploaded the revised manuscript, which contains the additional experiments and clarifications discussed in our rebuttal.
The key changes are summarized below:

1. Infrared Dataset for Missing Person Detection (Appendix Section G)
2. Zero-shot Evaluation with Vision-Language Models (Appendix Section H)
3. Evaluating SAR-Trained Models on a Real MAV Dataset (Appendix Section I)
4. Training Generative Models to Create Extreme SAR Situation (Appendix Section J)
5. Elucidating the Effect of Video Clip Length (Appendix Section K)
6. Real-Time Performance on Various Hardware Specifications (Appendix Section L)
7. Additional experiments investigating the impact of utilizing various metadata attributes during training (Appendix M)
8. Updated Ethics Statement for prohibiting non-humanitarian uses (Ethics Statement)

We believe these revisions comprehensively address the concerns raised. We kindly invite you to review the updated manuscript and consider these improvements in your final assessment.

Best regards,

The Authors

---

### Meta-Review · Area_Chair_4MfM · 2026-01-07

**Summary:**

This paper introduces ForestPersons, a high-quality and carefully constructed dataset for under-canopy missing person detection, which several reviewers agreed fills an important real-world gap. While concerns were raised about scope and venue fit, the dataset quality, detailed annotations, and the authors’ strong engagement during the rebuttal phase helped alleviate many technical and realism-related issues. Overall, this submission is on the borderline of acceptance, leaning positive due to its quality and the authors’ responsiveness. However, there remain potential ethical considerations around data usage, and input from the SAC and PC would be helpful to confirm whether any ethical issues remain.

**Reviewer Concerns:**

The rebuttal addressed several concrete concerns, including domain gap via real MAV data, RGB-only limitations through an infrared release plan, and zero-shot VLM evaluations, responding to KaHw, DXNw, and stPN. Additional analyses improved clarity and completeness. However, concerns from DXNw and FMJT about limited novelty, narrow scope, and ICLR suitability remain largely unresolved.

**Reviewer Scores:**

Reviewer stPN would likely remain positive, as their requests were thoroughly addressed and further experiments were added. Reviewer KaHw might move slightly upward from a borderline score after the real MAV evaluation and infrared clarification. Reviewer DXNw is unlikely to change the reject score, as concerns about novelty and scope persist. Reviewer FMJT would likely remain borderline, acknowledging the solid execution.

---

### Decision · Program_Chairs · 2026-01-26

Accept (Poster)